# B. subtilis MutS2 splits stalled ribosomes into subunits without mRNA cleavage

Esther N Park[1], Timur Mackens-Kiani[2], Rebekah Berhane[1], Hanna Esser[2], Chimeg Erdenebat[2], A Maxwell Burroughs[3], Otto Berninghausen[2], L Aravind [3], Roland Beckmann [2], Rachel Green [1,4] & Allen R Buskirk [1✉]

## Abstract

Stalled ribosomes are rescued by pathways that recycle the ribosome and target the nascent polypeptide for degradation. In *E. coli*, these pathways are triggered by ribosome collisions through the recruitment of SmrB, a nuclease that cleaves the mRNA. In *B. subtilis*, the related protein MutS2 was recently implicated in ribosome rescue. Here we show that MutS2 is recruited to collisions by its SMR and KOW domains, and we reveal the interaction of these domains with collided ribosomes by cryo-EM. Using a combination of in vivo and in vitro approaches, we show that MutS2 uses its ABC ATPase activity to split ribosomes, targeting the nascent peptide for degradation through the ribosome quality control pathway. However, unlike SmrB, which cleaves mRNA in *E. coli*, we see no evidence that MutS2 mediates mRNA cleavage or promotes ribosome rescue by tmRNA. These findings clarify the biochemical and cellular roles of MutS2 in ribosome rescue in *B. subtilis* and raise questions about how these pathways function differently in diverse bacteria.

**Keywords** Ribosome Quality Control (RQC); Ribosome Collisions; MutS2; *B. subtilis*; ABC ATPase
**Subject Categories** Microbiology, Virology & Host Pathogen Interaction; Structural Biology; Translation & Protein Quality

## Introduction

In bacteria, translating ribosomes stall when they encounter problems with an mRNA template, such as nucleotides that are chemically damaged and therefore unreadable, or truncations of the mRNA that result in the loss of the stop codon (Yan and Zaher, 2019; Thomas et al, 2020). Ribosomes also stall when elongation is slowed by low concentrations of aminoacyl-tRNA at clusters of rare codons or by specific peptide sequences that are difficult for the active site to accommodate (such as polyproline sequences) (Roche

and Sauer, 2001; Ude et al, 2013; Doerfel et al, 2013; Woolstenhulme et al, 2013). Indeed, certain arrest peptides such as SecM and TnaC take advantage of reversible ribosome stalling as a means to regulate the expression of downstream genes (Nakatogawa and Ito, 2002; Su et al, 2021; Gong and Yanofsky, 2002; Bhushan et al, 2011). In addition, bacterial ribosomes are arrested by many antibiotics (Polikanov et al, 2018; Wilson, 2009). If left unresolved, ribosome stalling by any of these mechanisms can be dangerous to the cell because it reduces the pool of active ribosomes and leads to the production of truncated, potentially toxic proteins.

Over the course of evolution, these problems imposed selective pressure that favored the emergence of dedicated pathways that rescue stalled ribosomes. These pathways accomplish the twin tasks of recovering the ribosomes and targeting the truncated nascent peptides and problematic mRNAs for degradation (Buskirk and Green, 2017). The best-characterized pathway in bacteria involves transfer-messenger RNA (tmRNA) which selectively rescues ribosomes stalled at the end of truncated mRNAs lacking a stop codon (so-called "non-stop" messages) (Ivanova et al, 2004; Neubauer et al, 2012). The ribosome resumes translation using tmRNA as a template, adding a peptide tag to the nascent polypeptide that targets it for degradation by proteases, primarily ClpXP (Keiler et al, 1996). Nearly all bacterial genomes encode tmRNA. There are also backup mechanisms that become engaged when tmRNA is overwhelmed. In *E. coli* and in *B. subtilis*, the backup pathway involves a small protein (ArfA/BrfA, respectively) that recruits a release factor to hydrolyze the nascent peptidyl-tRNA and promote recycling of the ribosome subunits without targeting the peptide for degradation (Garza-Sánchez et al, 2011; Chadani et al, 2010; Shimokawa-Chiba et al, 2019). Both of these pathways show a preference for ribosomes stalled on truncated mRNAs and require that the active site of the ribosome be competent to catalyze peptidyl transfer (for tmRNA) or peptidyl hydrolysis (for ArfA).

Several bacteria, including *B. subtilis*, also have a distinct pathway that shares similarities to the eukaryotic pathway known as ribosome-associated quality control (RQC). In eukaryotes, stalled ribosomes are split into subunits, yielding a free small subunit and a large subunit with a trapped peptidyl-tRNA (Matsuo et al, 2017, 2020). A factor called Rqc2 in yeast then catalyzes the addition of C-terminal Ala and Thr tails (CAT tails) to the nascent

[1]Department of Molecular Biology and Genetics, Johns Hopkins University School of Medicine, Baltimore, MD, USA. [2]Gene Center and Department of Biochemistry, University of Munich, Munich, Germany. [3]Computational Biology Branch, Intramural Research Program, National Library of Medicine, National Institutes of Health, Bethesda, MD, USA. [4]Howard Hughes Medical Institute, Johns Hopkins University School of Medicine, Baltimore, MD, USA. ✉E-mail: buskirk@jhmi.edu

peptide, translocating the peptide out of the tunnel such that encoded Lys residues can be tagged with ubiquitin by Ltn1 and ultimately degraded by the proteasome (Filbeck et al, 2022; Kostova et al, 2017; Shen et al, 2015). In a similar fashion, *B. subtilis* contains a homolog of Rqc2 called RqcH which binds to dissociated 50 S subunits with peptidyl-tRNA trapped on them and catalyzes the addition of Ala residues (Ala-tails) to the nascent peptide (Lytvynenko et al, 2019; Crowe-McAuliffe et al, 2021; Takada et al, 2021; Filbeck et al, 2021). Like the tmRNA tag, these Ala-tails target the nascent peptide for degradation by the bacterial proteasome equivalent, ClpXP. Many questions remain regarding how the RQC pathway operates in bacteria including: (1) what are the natural substrates of this pathway and how are they recognized, (2) how are stalled ribosomes split in order to generate the 50S-peptidyl-tRNA substrate for RqcH, and (3) how is the nascent peptide hydrolyzed from the tRNA and released.

We recently showed that ribosome rescue in *E. coli* is triggered by collisions that occur when a trailing ribosome catches up to a stalled ribosome (Saito et al, 2022). The stable interaction between the two ribosomes (primarily through their 30 S subunits) creates a new interface that recruits a factor called SmrB. This factor has an SMR domain that performs endonucleolytic cleavage of mRNAs when bound between collided ribosomes; cleavage occurs just upstream of the stalled ribosome. This cleavage in the ORF creates a nonstop mRNA such that upstream ribosomes that translate to this newly formed 3'-end are rapidly rescued by tmRNA. In addition to the cryo-EM structure of *E. coli* collided ribosomes bound to SmrB, we also reported the structure of collided ribosomes from *B. subtilis*, arguing that collisions are a conserved mechanism for recognizing stalled ribosomes in bacteria (Ferrin and Subramaniam, 2017), much like in yeast and human cells (Saito et al, 2022; Simms et al, 2017; Juszkiewicz et al, 2018; Ikeuchi et al, 2019).

Pfeffer and Joazeiro also reported the structure of collided ribosomes from *B. subtilis* bound to a factor homologous to SmrB called MutS2 (Cerullo et al, 2022). Like SmrB, MutS2 contains an SMR domain, but unlike SmrB it also contains several other domains including an ABC ATPase domain. The structure revealed that MutS2 binds to collided ribosomes as a dimer and that its ATPase domains contact the lead ribosome (Cerullo et al, 2022). These observations raised the exciting possibility that MutS2 recognizes collided ribosomes specifically and uses its ATPase domain to split the stalled ribosomes into subunits. Thus, MutS2 could be the missing factor required to dissociate ribosomes to promote Ala-tailing by RqcH. It remained unclear, however, how MutS2 selectively binds collided ribosomes since the ATPase domains bind to the lead ribosome alone and the SMR domain was not resolved in their structure. Furthermore, these studies did not establish whether MutS2 cleaves mRNA using its SMR domain as we had observed with *E. coli* SmrB (Cerullo et al, 2022).

Here, we thoroughly characterize the role of MutS2 in ribosome rescue in *B. subtilis*. We show that MutS2 is recruited by ribosome collisions and report the cryo-EM structure of the SMR and KOW domains of MutS2 bound to collided ribosomes. We find that the SMR domain plays an important role in recruiting MutS2 to collided ribosomes. Using a reporter construct in vivo, we show that MutS2 uses its ATPase activity to split ribosomes into subunits, promoting Ala-tailing of the nascent peptide by RqcH. Importantly, we see no evidence of mRNA cleavage by MutS2, arguing that it does not act upstream of the tmRNA pathway as SmrB does in *E.*

*coli*. Finally, we reconstitute the "rescue" reaction in vitro using purified collided ribosomes and show that MutS2 splits the stalled ribosomes into subunits in an ATP-dependent fashion but lacks detectable endonuclease activity. These findings define the biochemical activities of MutS2 in ribosome rescue in *B. subtilis*.

## Results

### Different architectures of bacterial SMR domain proteins

SMR domain proteins recognize ribosome collisions and cleave mRNA in *S. cerevisiae* (Cue2), *C. elegans* (NONU-1), and *E. coli* (SmrB) during ribosome rescue (Saito et al, 2022; D'Orazio et al, 2019; Glover et al, 2020). This study was prompted by our observation that SMR domain proteins are broadly conserved in bacteria and cluster in three major clades with distinct domain architectures (Fig. 1A,B). In *E. coli* and many other proteobacteria, the SMR domain is preceded by a relatively unstructured N-terminal extension, as observed in SmrB (21 kD). Our previous structural and biochemical studies revealed that conserved residues in a helix in this extension (the N-terminal hook) bind to the ribosomal protein uS2 and play a key role in recruiting SmrB to ribosomes (Saito et al, 2022).

In contrast, the architecture most commonly found in other bacterial phyla is more complex, typified by the MutS2 protein in *B. subtilis* (87 kD). From N- to C-terminus, this architecture contains the core/lever, clamp, P-loop ABC ATPase, KOW, and SMR domains (Fig. 1B). Notably, although the core/lever, clamp, and ATPase domains take their names from the homologous MutS protein involved in DNA mismatch repair (Lamers et al, 2000), the MutS2 architecture lacks two N-terminal domains found in MutS, the MutSI (mismatch recognition) and MutSII (connector) domains. The absence of these two domains argues against MutS2 being involved in mismatch repair.

Finally, the third clade is the smallest, restricted to the bacteroidetes lineage. SMR domain proteins in this clade only have an N-terminal KOW domain and C-terminal SMR domain (e.g., *C. baltica*, Fig. 1B), occasionally with an IG-like domain in between the two. Notably, some KOW domains in bacteria are known to have ribosome binding activity; for example, the KOW domain of NusG binds to ribosomal protein uS10 (Saxena et al, 2018), raising the possibility that the KOW domain in these two architectures (MutS2-like and KOW-SMR) plays a role in recruiting SMR domain proteins to the ribosome.

Alignment of the SMR domains also revealed that residues previously implicated in mRNA cleavage are not equally conserved in these three clades. The DxH and GxG motifs in the SMR domain play a role in mRNA cleavage and RNA binding in *E. coli* (Saito et al, 2022), yeast (D'Orazio et al, 2019), and plants (Zhou et al, 2017); these residues are highly conserved in SmrB proteins in proteobacteria and in the KOW-SMR protein in bacteroidetes. In contrast, we identified several independent occasions where the DxH active site residues have been wholly or partly lost in the MutS2 clade. An alignment of the SMR domain from MutS2 in firmicutes is shown in Fig. EV1A. Many proteins have completely lost the DxH motif, whereas others such as *B. subtilis* MutS2 have the residues DLR which do not fully conform to the consensus DxH motif. In contrast, the GxG motif is highly conserved in firmicutes,

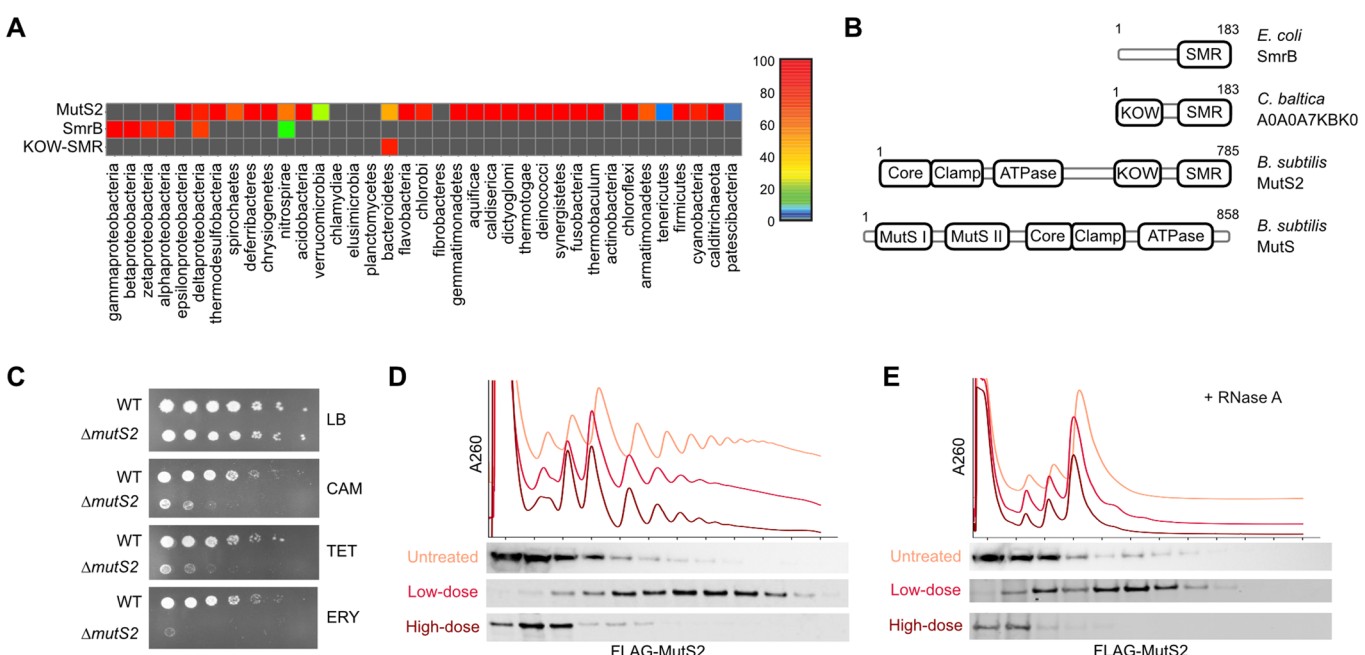

**Figure 1.    MutS2, an SMR domain protein in *B. subtilis*, binds collided ribosomes.**

(A) Heatmap showing the percentage of genomes in each bacterial phylum encoding an SMR domain protein. (B) Domain organization of three representative bacterial SMR domain proteins and the related DNA mismatch repair factor MutS. (C) Spotting assay showing Δ*mutS2* cells are hypersensitive to chloramphenicol (CAM) (1 μg/ mL), tetracycline (TET) (2 μg/mL), and erythromycin (ERY) (0.08 μg/mL). (D) Low doses of CAM induce collisions whereas high doses stall ribosomes without inducing collisions. Following treatment with low-dose (2 μg/mL) and high-dose (200 μg/mL) CAM, the distribution of FLAG-MutS2 was determined by fractionation over sucrose gradients and detection with an anti-FLAG antibody. (E) Lysates from cells with and without CAM were treated with RNase A, fractionated over sucrose gradients, and the binding of FLAG-MutS2 to nuclease-resistant disomes was detected with an anti-FLAG antibody. Source data are available online for this figure.

as is the His residue just upstream (residue His743 in *B. subtilis* MutS2). These observations raise questions about whether SMR domains in the MutS2 clade retain the endonucleolytic activity observed in *E. coli* SmrB.

## Δ*mutS2* cells are hypersensitive to antibiotics that target ribosomes

To explore whether the MutS2 protein in *B. subtilis* plays a role in translation, we first examined the phenotype of a strain lacking this factor. Δ*mutS2* cells did not have a significant growth defect compared to wild-type cells on plates made with rich medium. However, cells lacking MutS2 are hypersensitive to several antibiotics that target the ribosome. On plates with chloramphenicol (CAM), tetracycline (TET), or erythromycin (ERY), the growth of the Δ*mutS2* strain is less robust than wild-type (Fig. 1C). In contrast, the Δ*mutS2* strain is not sensitive to beta-lactam antibiotics (e.g., carbenicillin) that target cell wall synthesis (Appendix Fig. S1). These results suggest that MutS2 plays a role in mediating the toxicity of antibiotics that perturb the elongation stage of protein synthesis.

## MutS2 preferentially binds collided ribosomes

We next asked whether MutS2 associates with ribosomes in vivo. To facilitate the detection of MutS2, we ectopically expressed an

N-terminally FLAG-tagged MutS2 construct from its native promoter in the Δ*mutS2* strain. We treated these cells with varying concentrations of CAM to ask how ribosome collisions affect MutS2 binding to ribosomes. As shown previously in yeast and *E. coli*, high concentrations of antibiotics that target the ribosome stall ribosomes quickly in place, whereas lower doses only stall some ribosomes, allowing others to continue elongating until they collide with the stalled ribosomes (Saito et al, 2022; Simms et al, 2017). We used this strategy to ask if MutS2 binds preferentially to collided ribosomes. In the untreated sample, MutS2 mostly is found in the light fractions of the sucrose gradient, although some portion is also found associated with monosomes and light polysomes, arguing that it can bind to ribosomes generally (Fig. 1D). MutS2 is enriched in polysomes deeper in the gradient when cells were treated with a low dose of CAM (2 μg/mL), a concentration roughly equivalent to the minimum inhibitory concentration (MIC). Importantly, the enrichment of MutS2 in polysomes is lost in cells treated with much higher concentrations of CAM (200 μg/mL) (Fig. 1D). We conclude that MutS2 weakly binds ribosomes in general and that its binding is enhanced by collisions, not merely by ribosome stalling.

We also asked whether MutS2 is preferentially recruited to nuclease-resistant disomes, a hallmark of collided ribosomes. Treatment of lysates with RNase A typically collapses most polysomes into monosomes, but when ribosomes have collided, RNase A cannot access the mRNA in the tight interface between them, thus leading to disome accumulation (Juszkiewicz et al,

2018). In untreated samples, polysomes collapsed into monosomes and MutS2 was mostly present in the lighter fractions (Fig. 1E). However, in cells treated with 2 µg/mL CAM to induce collisions, small peaks corresponding to nuclease-resistant disomes and trisomes were identified; we observe that MutS2 is strongly enriched in those deeper fractions (Fig. 1E). As expected, in samples treated with high concentrations of CAM, MutS2 was not enriched in the heavier fractions.

## The structure of the KOW and SMR domains of MutS2 on collided ribosomes

The structure of MutS2 as a homodimer bound to collided ribosomes was previously visualized by cryo-electron microscopy by Cerullo et al. Although their structure reveals the overall arrangement of the lever, clamp, and ATPase domains of MutS2 on collided disomes, it does not provide insight into the positioning of the KOW and SMR domains in this interaction, nor does it reveal how collided ribosomes are specifically recognized. In order to further elucidate the mechanisms of MutS2 recruitment and activity, we reconstituted the complex in vitro. From an in vitro translation reaction of the MifM stalling construct in *B. subtilis* lysates, we purified disomes from a sucrose density gradient as described previously (Saito et al, 2022; Chiba and Ito, 2012). We purified recombinant *B. subtilis* MutS2 (Fig. EV2A) and observed by size-exclusion chromatography that it exists as an oligomer (Fig. EV2B) as reported for MutS2 from other bacteria (Damke et al, 2016; Fukui et al, 2007). A tenfold excess of MutS2 protein was added to the disomes and the reaction was incubated in the presence of AMP-PNP. The sample was then vitrified and subjected to cryo-EM and single-particle analysis.

We observe two major classes of collided disomes: one with only mRNA density in the interribosomal space and another with an additional density next to the mRNA (Figs. 2A,B and EV3). The latter class also contains additional density next to uS10 on both the stalled and the collided ribosomes. By local refinement of the experimental data and rigid body-fitting of a model of MutS2 generated in Alphafold2 (Jumper et al, 2021), we identified these extra densities as the SMR and KOW domains, respectively (Fig. 2B). As described below, we were not able to visualize the N-terminal domains seen in the previous structure (lever, clamp, and ATPase).

The overall architecture of the collided disome bound by the MutS2 KOW and SMR domains is shown in Fig. 2C. Only one SMR domain is visible, next to the mRNA in the inter-ribosomal space, interacting with both ribosomes (Fig. 2D). In this position, the region of the SMR domain containing the DLR and GxG residues is oriented towards the mRNA, suggesting a specific interaction as we previously observed for SmrB in *E. coli* (Fig. 2F,G) (Saito et al, 2022). On the stalled ribosome, the SMR domain interacts with uS11 and uS7, as does SmrB; on the collided ribosome, the SMR domain is positioned next to uS3.

The two KOW domains bind to ribosome protein uS10 in a manner that is highly similar to the KOW domain of NusG, an *E. coli* protein involved in transcription-translation coupling (Saxena et al, 2018). The KOW domain on the collided ribosome is connected to the SMR domain by partial density, consistent with a flexible loop joining them, arguing that these domains derive from the same MutS2 monomer. The KOW domain on the stalled ribosome is part of the second MutS2 in the homodimer whose other domains are not resolved. The position of the two KOW domains is compatible with the position of the coiled-coil domains in MutS2 in the previous structure (Cerullo et al, 2022). Indeed, partial density for the KOW domain on the collided ribosome is seen linked to the N-terminal domains in the electron density maps of the earlier study though it was not discussed there (Cerullo et al, 2022). We conclude that no structural rearrangements would be required to link the density of the N-terminal domains in the previous structure with the C-terminal KOW and SMR domains reported here (Fig. EV4A–C).

Finally, although the SMR domains of both MutS2 and SmrB recognize composite binding sites formed between the collided ribosomes near the bridging mRNA, the orientation of the SMR domain is very different in the two complexes from *B. subtilis* and *E. coli*. This difference in the SMR domain orientation may arise from constraints imposed by additional interactions of the N-terminal hook of SmrB with uS2 and by the MutS2 KOW domain with uS10. As a result, compared to SmrB, the SMR domain of MutS2 is rotated around the mRNA by ~120° (Fig. 2E). Together with the lack of strong amino acid conservation, this finding raises the question as to whether the SMR domain of MutS2 possesses the same endonucleolytic activity as SmrB (Saito et al, 2022).

## Attempts to reconcile the differences in the structures

Not only were different MutS2 domains resolved in our structure and the previous one by Cerullo et al, there are also discrepancies in the conformation and composition of the ribosome. Here, only in disomes where both ribosomes adopt a non-rotated (or classical) conformation do we find density corresponding to the SMR domain, despite the presence of collided disomes in other states in our dataset. In contrast, in the Cerullo structure, the collided ribosome appears in a rotated (or hybrid) conformation. In addition, there are differences in the ribosome proteins present at the disome interface: bS21 is missing in the stalled ribosome in our structure (Fig. EV4F,G) where it was previously observed, and uS2 is clearly present on both the stalled and collided ribosomes in our structure (Figs. 2C and EV4E), although it was previously missing from the stalled ribosome.

Thinking that these discrepancies might arise from differences in how the complexes were prepared, we isolated native MutS2-disome complexes from *B. subtilis* cells expressing FLAG-tagged MutS2 (containing the Walker B mutation) by affinity purification. Analysis of this native complex revealed a cryo-EM reconstruction containing extra densities corresponding to the SMR and KOW domains (Fig. EV4H), albeit at a lower resolution than the in vitro complex described above, but without density corresponding to the N-terminal domains. Likewise, we observed clear density for uS2 on the stalled ribosome and at most partial density in the binding pocket of bS21 (Fig. EV4I). The collided ribosome, however, assumes the same rotated conformation described previously (Cerullo et al, 2022), meaning that binding of the SMR domain is compatible with this ribosome conformation. We speculate that MutS2 may bind collided ribosomes in a non-rotated state (as we observed in vitro) followed by a transition to a rotated state in vivo or during the purification procedure. These experiments on the native complex suggest that the discrepancies in the ribosome

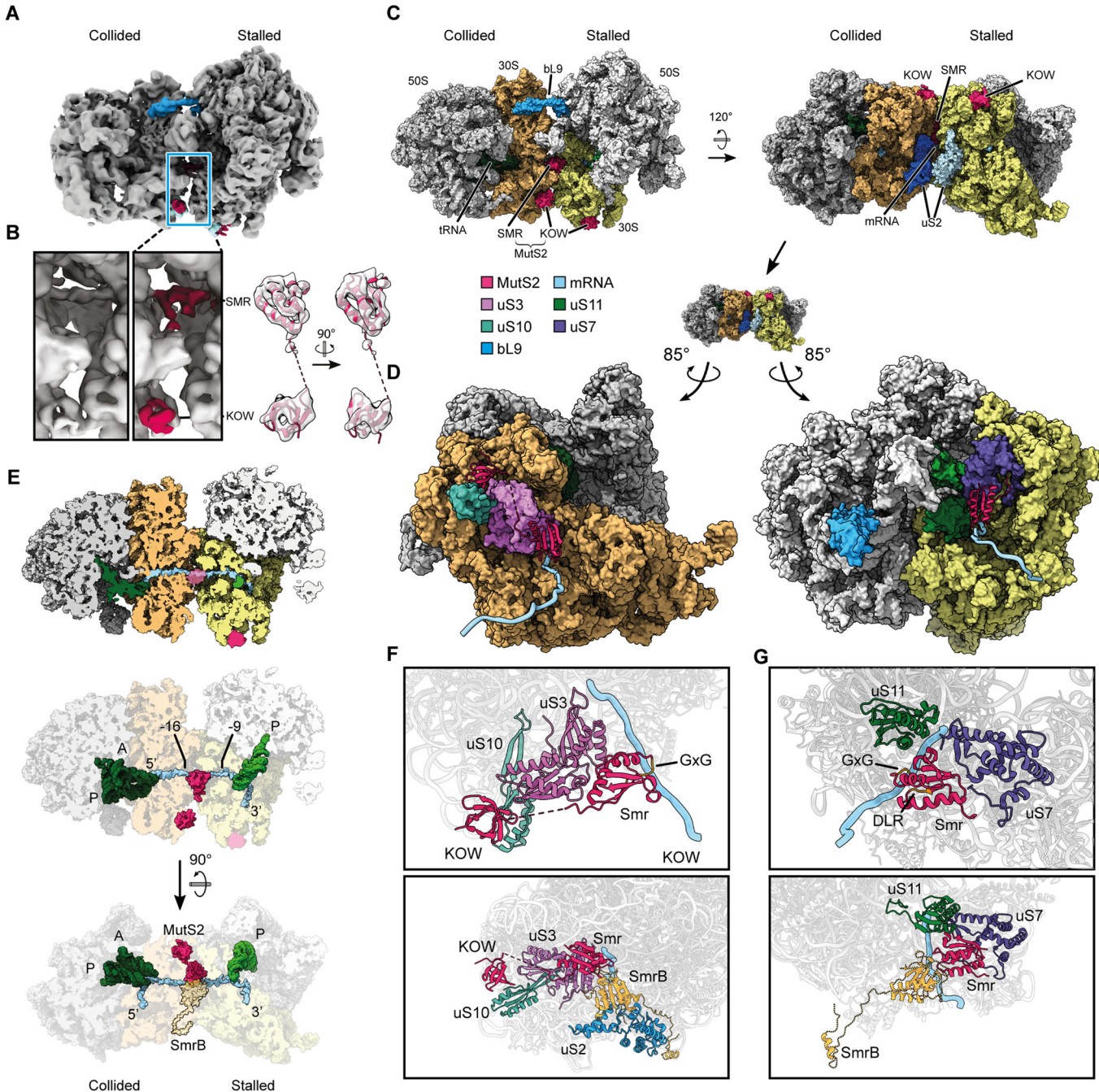

**Figure 2. Cryo-EM structure of the MutS2 KOW and SMR domains binding the *B. subtilis* disome.**

(A) Experimental cryo-EM reconstruction, lowpass-filtered, with MutS2 in red, uS10 in light blue and bL9 in blue. (B) Left: Zoomed-in view of the inter-ribosomal interface of a class of particles without (left) and with (right) MutS2. Right: Fit of the MutS2 KOW and SMR domains into the experimental density. (C) Cryo-EM structure of the collided *B. subtilis* disome bound by MutS2 KOW and SMR domains. (D) Interactions of MutS2 with the collided disome interface, seen from each side of that interface. (E) Top, Middle: Cut view of the MutS2-bound disome showing the mRNA path and the position of the KOW and SMR domains as well as tRNA (green) in both ribosomes. Bottom: Comparison of the position of MutS2 in the *B. subtilis* disome to that of SmrB in the *E. coli* disome. (F) Top: Close-up view of the conformation and interactions of MutS2 on the collided ribosome side of the interface. Bottom: Overlay of the same with the *E. coli* SmrB structure from Saito et al (matched to the *B. subtilis* structure by aligning uS2 from both structures). (G) Top: Close-up view of the conformation and interactions of MutS2 on the stalled ribosome side of the interface. Bottom: Overlay of the same with the *E. coli* SmrB structure from Saito et al.

protein composition and MutS2 structures reported here and by Cerullo et al are not due to the formation of the complex in vivo or in vitro.

One interpretation is that MutS2 interacts with collided disomes in two states, either through the KOW and the SMR domains or through the KOW and N-terminal domains (lever, clamp, ATPase). Perhaps the SMR and KOW domains facilitate disome recognition and MutS2 recruitment, and when the ATPase domain binds in a pre-splitting conformation, the interactions with the SMR domain become dispensable and thus no longer visible by cryo-EM.

## MutS2 releases truncated proteins from stalled ribosomes but does not affect mRNA levels

To study the activity of MutS2 in vivo, we designed reporter constructs that allow us to follow the translation of a problematic mRNA in *B. subtilis* (Fig. 3A). Each reporter contains an in-frame fusion of NanoLuc to the bleomycin resistance protein (BleR). We created two control constructs, one with a stop codon between the genes that produces NanoLuc alone (Stop) and a second without any stalling motif (Non-stall) that produces the full-length fusion protein. In a third construct, we inserted the 31-residue ApdA stalling motif between NanoLuc and BleR (ApdA). This arrest peptide from *A. japonica* arrests elongating *B. subtilis* ribosomes by inhibiting peptidyl transfer (Sakiyama et al, 2021).

To confirm that ribosome stalling at ApdA triggers downstream rescue pathways, we performed a western blot using antibodies against NanoLuc. A strong band corresponding to full-length protein is observed for the Non-stall control and loss of MutS2 did not affect this reporter, as expected (Fig. 3B). In contrast, there is significantly less full-length protein for the ApdA reporter because stalling lowers the protein output. Moreover, the ApdA reporter generates a truncated protein that is slightly larger than the NanoLuc produced from the Stop control, consistent with the translation of the additional ApdA sequence prior to ribosome stalling. Importantly, the loss of MutS2 resulted in a substantial decrease in the amount of truncated reporter protein from the ApdA reporter (Fig. 3B). These results are consistent with a model in which MutS2 rescues ribosomes stalled in the middle of an open reading frame (ORF) thus releasing truncated protein products.

In addition, we analyzed the activity of MutS2 on the reporter mRNA using northern probes specific for the 5'- or 3'-ends (Fig. 3C). With the 5'-probe, we see primarily full-length mRNA from the Non-stall reporter. Remarkably, there appear to be similar levels of full-length mRNA from the ApdA reporter as well, in stark contrast to our previous observation in *E. coli* that the presence of a strong arrest peptide dramatically lowers full-length reporter mRNA levels (Saito et al, 2022). This finding suggests that unlike in *E. coli*, where ribosome stalling targets transcripts for decay by SmrB nuclease activity, ribosome stalling does not target the reporter mRNA for decay in *B. subtilis*. With the 3'-probe, again we see that full-length mRNA levels are similar with or without the ApdA stalling sequence. With this probe, we also detect a decay intermediate from the ApdA reporter corresponding to the mRNA fragment downstream of the stall site; importantly, loss of MutS2 does not affect the level of this fragment, suggesting that MutS2 is not responsible for its production. We speculate that the truncated band arises from the degradation of the upstream mRNA by 5'–3' exonucleases until they are blocked by the stalled ribosome,

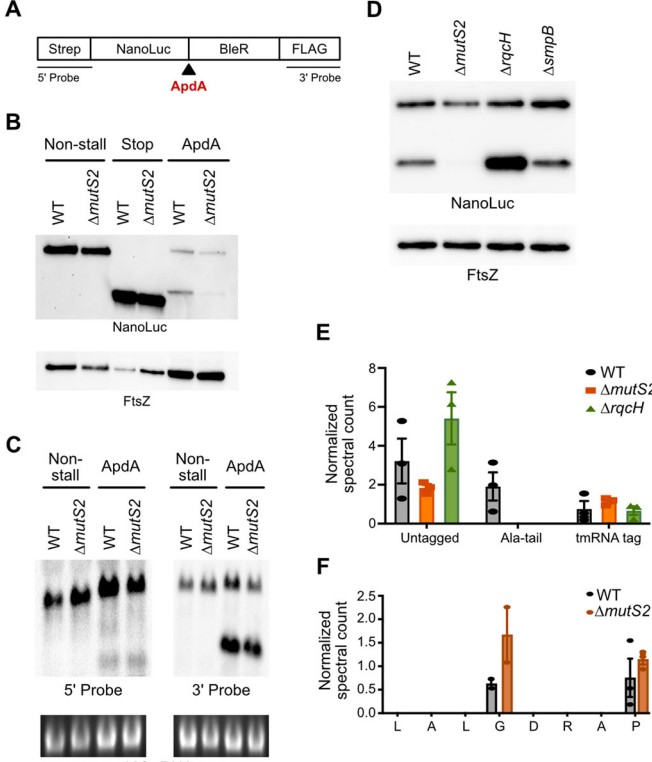

**Figure 3. MutS2 rescues ribosomes stalled in the middle of an ORF.**

(A) Schematic of stalling reporter for studying ribosome rescue in *B. subtilis*. Between the NanoLuc gene and bleomycin resistance gene, we inserted either no additional sequence (Non-stall), stop codons (Stop), or the ApdA stalling motif. (B) Reporter protein from wild-type and Δ*mutS2* strains was detected by anti-NanoLuc antibodies. The FtsZ protein serves as a loading control. (C) Northern blots of reporter mRNA using the 5'-probe and the 3'-probe. Ethidium bromide staining of the 16 S rRNA serves as a loading control. (D) Reporter protein from wild-type, Δ*mutS2*, Δ*rqcH*, and Δ*smpB* strains was detected by anti-NanoLuc antibodies. The FtsZ protein serves as a loading control. (E) The addition of the tmRNA tag or the Ala-tail at the stall site of the reporter in wild-type, Δ*mutS2*, and Δ*rqcH* strains was detected by LC-MS/MS. The spectral counts are normalized to a different peptide from the reporter protein. The mean of three biological replicates and the standard error are shown. (F) tmRNA tagging levels along the stalling sequence in wild-type and Δ*mutS2* strains are not dependent on MutS2. The mean of three biological replicates and the standard error are shown. Source data are available online for this figure.

yielding a stable, downstream fragment (Trinquier et al, 2020; Mathy et al, 2007). Taken together, these data show that MutS2 generates a truncated protein product from ribosomes stalled in the middle of an ORF but does not affect mRNA levels, suggesting that, unlike SmrB, it may indeed lack nuclease activity.

## Ala-tailing by RqcH depends on MutS2

We envisioned that MutS2 might release truncated proteins from stalled ribosomes in two different ways, depending on the activity of the protein; these possibilities are not mutually exclusive. First, if MutS2 were to cleave the mRNA on collided ribosomes, then upstream ribosomes would arrest at the newly formed 3'-end and be rescued by tmRNA, leading to the release of a truncated protein

with C-terminal SsrA tags encoded by tmRNA during the rescue process. Typically, SsrA-tagged proteins are rapidly degraded by the ClpXP protease (Fei et al, 2020). If the tmRNA system is overwhelmed, however, a backup system involving ArfA in *E. coli* or BrfA in *B. subtilis* typically releases the nascent peptide without adding a degron tag (Garza-Sánchez et al, 2011; Chadani et al, 2010; Shimokawa-Chiba et al, 2019). In *E. coli*, we previously observed that cells lacking tmRNA generate far higher levels of truncated protein products from a stalling reporter because the ArfA-released (and therefore untagged) protein products are stable relative to those that were tagged by tmRNA (Saito et al, 2022). We find that in *B. subtilis* cells in which the tmRNA pathway was inactivated by deletion of its protein partner SmpB, there is no difference in the level of truncated protein produced compared to the wild-type cells (Fig. 3D); thus, there is no major role for the tmRNA system in resolving the stalled ribosomes on the ApdA reporter. This finding is consistent with the lack of evidence that MutS2 cleaves the reporter mRNA to generate a prototypical nonstop message substrate for tmRNA/SmpB.

A second possible mechanism of action is that the ATPase domain of MutS2 splits the stalled ribosome into subunits, freeing the 30 S subunit but yielding a 50 S subunit with peptidyl-tRNA trapped on it, akin to the activity of RQT in eukaryotic systems (Matsuo et al, 2017). It has been shown that in *B. subtilis* (as in eukaryotes) the 50S-peptidyl-tRNA complex is a substrate for the RQC factor RqcH which adds several Ala residues to the C-terminus of the nascent polypeptide and targets the protein for degradation by ClpXP (Lytvynenko et al, 2019; Crowe-McAuliffe et al, 2021; Takada et al, 2021; Filbeck et al, 2021). In this case, the expectation would be that deletion of RqcH should stabilize truncated proteins produced by this pathway because they would not be Ala-tailed (Lytvynenko et al, 2019). Indeed, we observe higher levels of truncated protein from the reporter construct in the absence of RqcH (Fig. 3D), consistent with the model proposed by Cerullo et al in which MutS2 splits ribosomes into subunits that are then acted on by RqcH to target the nascent peptide for degradation (Cerullo et al, 2022).

To further characterize the truncated protein, we used mass spectrometry to identify C-terminal fragments to detect whether SsrA-tag or Ala-tails were added during the rescue process. We grew cells with bortezomib (an inhibitor of ClpXP) to prevent degradation of the truncated proteins and immunoprecipitated the ApdA reporter protein from wild-type, *ΔmutS2*, and *ΔrqcH* strains. We then digested the protein with lysyl endopeptidase and subjected the resulting peptides to liquid chromatography with tandem mass spectrometry (LC-MS/MS). Ribosomes stall near the end of the ApdA sequence at the RAPP motif with the first Pro codon in the P site and the second Pro codon in the A site (Sakiyama et al, 2021). We observed abundant peptides in all three strains ending in RAP (Fig. 3E). These represent proteins unmodified by tmRNA or RqcH possibly arising from peptides released from the 50 S after splitting (without Ala-tails) or nascent peptides on 70 S ribosomes released from tRNA during sample preparation. More interestingly, we observe peptides with alanine tails added at the site of the stall (after RAP). These peptides are only observed in the wild-type strain. Deletion of RqcH leads to loss of Ala-tailing, as expected, and likewise, deletion of MutS2 also leads to loss of Ala-tailing (Fig. 3E). These findings are consistent with a model wherein MutS2 activity is upstream of Ala-tailing by

RqcH in vivo. Indeed, based on their observations of loss of Ala-tagging in strains lacking MutS2, Cerullo et al proposed renaming MutS2 as RqcU (RQC-upstream factor).

We also observed proteins that had been tagged by tmRNA. These peptides were less abundant than those released at the stall site (ending in RAP) or those with Ala-tails, although with the challenges in mass spectrometry in detecting various peptides we cannot make any strong quantitative conclusions. The tmRNA tag is added to peptides right at the stall site after the RAP motif (Fig. 3F) as well as at a second site after the Gly residue four residues upstream. In both cases, the number of tmRNA-tagged peptides was not reduced in samples prepared from cells lacking MutS2, arguing that MutS2 is not functioning upstream of tmRNA tagging at either site. We argue that tmRNA tagging arises from mRNA decay pathways unrelated to MutS2 in *B. subtilis*. These data are in stark contrast to our earlier observations in *E. coli* where tmRNA-tagged products arising from upstream of the stall site disappear when SmrB is deleted (Saito et al, 2022). Taken together, these data are consistent with a role for MutS2 in splitting the downstream ribosome into subunits (leading to Ala-tailing by RqcH) but do not provide evidence in support of mRNA cleavage by MutS2.

## The KOW and SMR domains promote MutS2 binding to collided ribosomes

With insights from the cryo-EM structure, we next made a series of MutS2 mutations to determine how each domain contributes to binding collided ribosomes in vivo. Mutant FLAG-tagged MutS2 constructs were expressed from an ectopic site in the *ΔmutS2* strain; the expression levels of all the mutants were found to be roughly equivalent (Appendix Fig. S2). We treated cultures with a low dose of CAM to induce collisions and performed western blots using anti-FLAG antibodies to follow MutS2 sedimentation across sucrose gradients.

Given that our structure revealed that the KOW domain binds to ribosomal protein uS10, we made several Ala mutations in a single construct designed to perturb the binding interface (KOW-mut), including Q668A, I671A, L672A, and K673A. These surface-exposed residues lie in a loop corresponding to the conserved F165 in the KOW domain of NusG that is critical for ribosome binding (Saxena et al, 2018). As expected, binding of the KOW-mut protein is reduced compared to the wild-type (Fig. 4A); there is more protein in the first fractions and less bound to ribosomes deeper in the gradient.

The fact that the SMR domain is buried between the collided ribosomes suggests that it may specifically sense collisions through the recognition of a distinct, composite binding interface. Indeed, deletion of the SMR domain (ΔSMR) through truncation after residue 701 dramatically reduces MutS2 binding to ribosomes (Fig. 4A). We also made mutations to conserved residues in the SMR domain likely to be involved in RNA binding, independently changing $D_{711}LR$ to ALA and mutating the conserved His just upstream of the GxG motif (H743A). Both the ALA mutant and the H743A mutant show dramatic reductions in binding to colliding ribosomes (Fig. 4A). In contrast, we found that mutation of the Walker B motif (E416A) in the ATPase domain had little or no effect; this mutant protein still shifts deep into polysomes when collisions are induced (Fig. 4A). These results with the MutS2

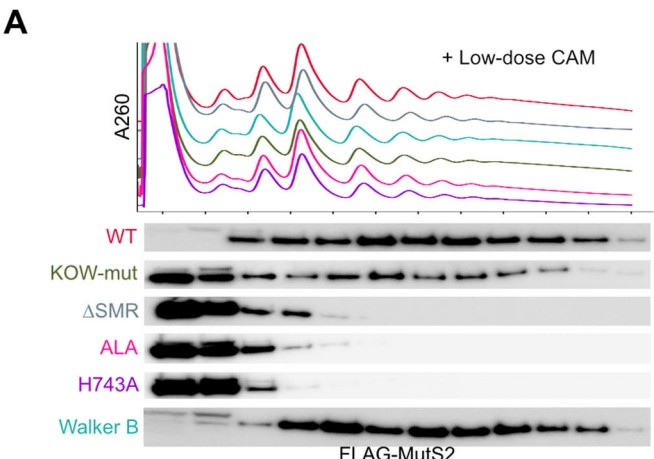

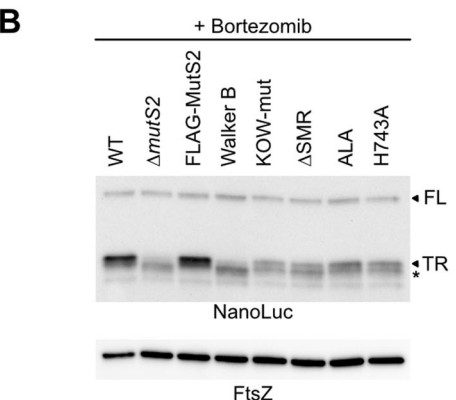

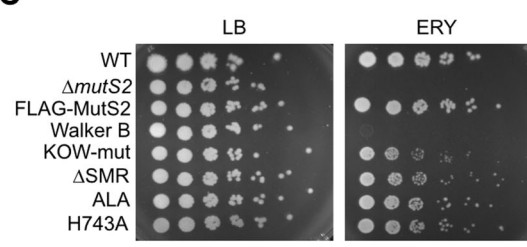

**Figure 4. Activities of the SMR and ATPase domains of MutS2.**

(A) Following induction of ribosome collisions with CAM, the distribution of Flag-tagged MutS2 and several mutants was determined by fractionation over sucrose gradients and detection with anti-FLAG antibody. ΔSMR is missing the SMR domain (residues 710–785). The Walker B mutant prevents ATP hydrolysis (E416A). KOW-mut contains mutations to the KOW domain to perturb binding to uS10 ($Q_{668}$xxILK to $A_{668}$xxAAA). ALA and H743A are mutations to conserved residues in the SMR domain. (B) Cells expressing various constructs of MutS2 were grown with 20 μM bortezomib to inhibit ClpXP activity and reporter protein was visualized using an anti-NanoLuc antibody. FL = full-length ApdA reporter protein, TR = truncated reporter protein, * = smaller truncated protein not dependent on MutS2 activity. The FtsZ protein serves as a loading control. (C) Spotting assay of strains expressing various MutS2 constructs on plates with and without erythromycin (0.08 μg/mL). Source data are available online for this figure.

mutants show that, consistent with our cryo-EM structure, the KOW and especially the SMR domain promote MutS2 binding to collided ribosomes.

## The ABC ATPase domain is critical for MutS2 function

To determine the effect of these mutations on the activity of MutS2, we introduced the ApdA stalling reporter into strains carrying the MutS2 mutants. We added bortezomib to cultures to prevent degradation of the truncated protein and performed western blots against NanoLuc, the upstream part of the stalling reporter. In wild-type cells, there is a strong band corresponding to the truncated reporter that is stabilized relative to the full-length protein by the addition of bortezomib (Fig. 4B) compared to untreated samples (Fig. 3B). There is also a band just below the major band that is not dependent on MutS2; in the *ΔmutS2* strain, only the top band decreases (TR), not the lower band (*). When a Flag-tagged copy of wild-type MutS2 is added to complement the deletion, the upper band, TR, is rescued to wild-type levels, indicating that the Flag-tag does not impact MutS2 activity.

The Walker B mutant yields little or no truncated MutS2 product (the TR band), similar to the complete knockout strain, *ΔmutS2* (Fig. 4B); these data indicate that inhibiting ATP hydrolysis abrogates MutS2 activity. In contrast, mutation of the KOW domain (KOW-mut) or the SMR domain (ΔSMR, ALA, H743A) yielded an intermediate phenotype, where we saw some reduction in the level of the upper band, but not a complete loss of MutS2 product. This loss of activity is likely due to the reduction in binding observed in Fig. 4A.

We also tested the effects of MutS2 mutations on the ability of cells to survive on plates with the collision-inducing antibiotic erythromycin (ERY). Just as *B. subtilis* cells lacking MutS2 are sensitive to ERY, so too are cells expressing the Walker B mutant (Fig. 4C). In contrast, cells expressing the KOW-mut or mutations in the SMR domain showed only very modest sensitivity to ERY. These results show that the ATPase domain of MutS2 is associated with its most critical functional domain.

## MutS2 splits disomes into ribosome subunits in vitro

We reconstituted disome splitting in vitro, purifying *B. subtilis* disomes from an in vitro translation reaction and combining them with purified wild-type or mutant MutS2. The reactions were then fractionated on a sucrose gradient in order to analyze the relative abundance of the remaining disomes, monosomes, and ribosome subunits, in order to determine the splitting efficiency.

When incubating the collided disomes with wild-type MutS2 in the presence of ATP, we generally observed a marked decrease in the disome peak compared to a control experiment in the absence of ATP. At the same time, we observed an increase in peaks corresponding to ribosomal subunits and monosomes, indicating disome splitting activity by MutS2 (Fig. 5A, red). In these experiments, the contribution of the disome peak area to the total for all ribosomal fractions decreased by at least 40%. However, when ATP was substituted with the non-hydrolyzable analog AMP-PNP, no such decrease was observed, confirming that ATP hydrolysis is required for MutS2 splitting activity and that ATP binding alone is not sufficient (Fig. 5A). We also found that the Walker A and Walker B mutants of MutS2 fail to split the disomes, even in the presence of ATP (Fig. 5B); these data establish that ATP binding followed by hydrolysis by the MutS2 ATPase domain is required for efficient dissociation of disomes in vitro. In contrast, repeating the experiment with the MutS2 ALA mutant, which

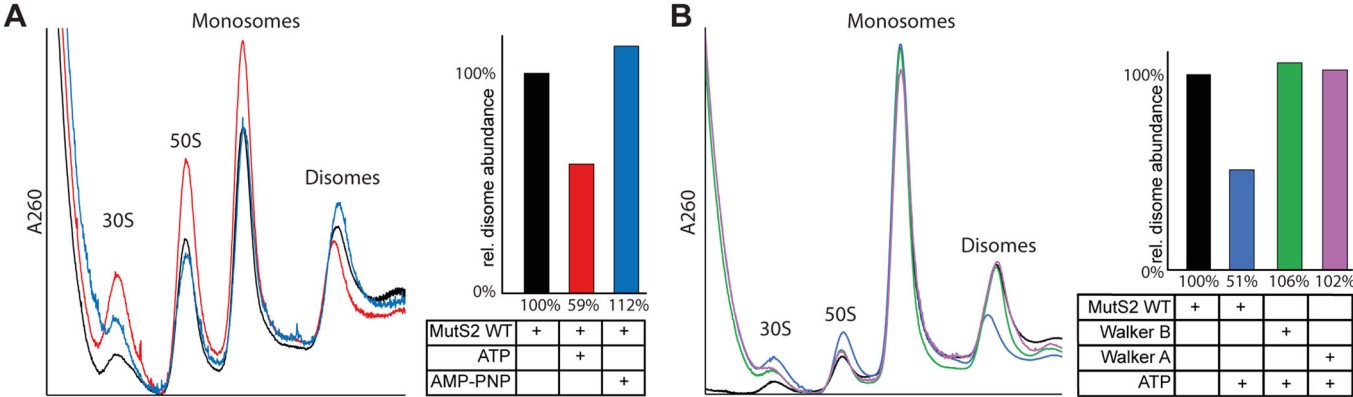

**Figure 5. MutS2 splits stalled ribosomes into subunits in vitro.**

(A) Left: UV chromatograms from sucrose gradient fractionation of disome splitting assays with MutS2 WT. Right: Relative abundance of disomes compared to total ribosomal fractions after splitting reaction in experiments with MutS2 WT, calculated from relative peak areas in the chromatograms by dividing the disome peak area by the total peak area of subunits, monosomes, and disomes. Purified *B. subtilis* disomes were used as input. Only in the presence of ATP do we observe a significant decrease in the abundance of disomes compared to other ribosomal fractions after incubation with MutS2. (B) Left: UV chromatograms from sucrose gradient fractionation of disome splitting assays with MutS2 WT and ATPase ("Walker B": E416A, "Walker A": G340R) mutants. Right: Relative abundance of disomes computed as above. Purified *B. subtilis* disomes were used as input. Mutations that render either the Walker A or Walker B motifs non-functional abrogate the disome splitting activity of MutS2 entirely. Source data are available online for this figure.

disrupts the DLR motif of the SMR domain, yielded similar splitting activity when comparing the SMR domain mutant with wild-type MutS2 (Fig. EV5). These data further argue that these residues in the SMR domain of MutS2 are not essential for disome splitting and that MutS2 does not carry out an endonuclease function.

## Discussion

The data presented here support a model in which *B. subtilis* MutS2 promotes the rescue of stalled ribosomes in a dramatically different manner from *E. coli* SmrB (Fig. 6). Although both proteins contain an SMR domain that recognizes the interface formed by two colliding ribosomes, helping to recruit them to their disome substrate, the biochemical activities of the two proteins are distinct. The SMR domain in SmrB functions as an endonuclease, cleaving mRNA between the collided ribosomes, allowing upstream ribosomes to translate to the newly formed 3'-end, where they are rapidly rescued by tmRNA. After canonical release and recycling on the tmRNA template, the tag encoded by tmRNA leads to degradation of the nascent peptide by proteases. In contrast, the SMR domain in *B. subtilis* MutS2 is not an active nuclease, nor does it target ribosomes for rescue by tmRNA. Instead, the ATPase domain of MutS2 splits the stalled ribosomes into subunits, freeing the 30 S subunit as well as a 50 S subunit bound to peptidyl-tRNA. RqcH then facilitates the non-templated addition of Ala residues to the C-terminus of the nascent peptide, and after the peptide is released from the tRNA through an unknown mechanism, the Ala-tail targets it for degradation by proteases. Through these distinct mechanisms, both SmrB and MutS2 trigger pathways that recycle the stalled ribosomes and degrade the aborted nascent polypeptides.

In this study, we clarify MutS2's mechanism of action in recognizing collided ribosomes in *B. subtilis*. Ribosome collisions

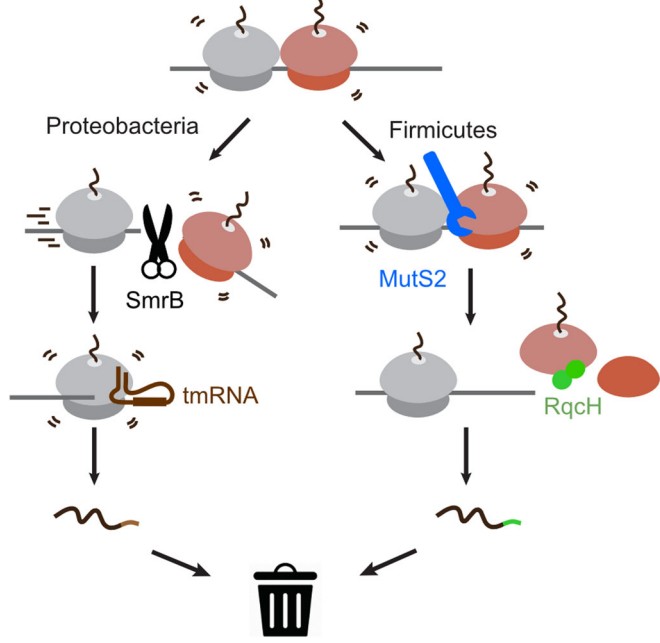

**Figure 6. Model for ribosome rescue in bacteria.**

Proteobacteria containing SmrB rescue collided ribosomes via nucleolytic cleavage while firmicutes and other bacteria containing MutS2 split collided ribosomes into subunits. These differences mean that different pathways (tmRNA or RqcH) tag the nascent polypeptide to target it for degradation by proteases.

are present in diverse bacteria and share common features. In *E. coli* and in *B. subtilis*, the SMR domain plays a role in recruiting both SmrB and MutS2 to collided ribosomes, recognizing the similar composite binding site formed between the two ribosomes. In both cases, residues in the DxH/DLR and HGxG

motifs are oriented towards the mRNA. In the case of SmrB, the DLH residues are involved in catalysis; in the case of MutS2, our data suggest that the DLR and HGxG sequences are required for high affinity binding to ribosomes but not for endonucleolytic cleavage. We note that the sucrose gradient sedimentation binding assay is a stringent test as evidenced by the fact that SMR domain mutants that fail to bind robustly still retain partial rescue activity. Ribosome binding is likely aided by auxiliary interactions of SmrB and MutS2 with the periphery of the collision interface, at sites that are accessible on all ribosomes, not only collided ones. For example, the interactions between the KOW domain of MutS2 and uS10 may be sufficient for partial activity even without the SMR domain. Most notably, as revealed by cryo-EM structures, the orientations of the SMR domains of SmrB and MutS2 are completely different, consistent with the difference in terms of catalytic activity of the two proteins.

Apparently, although SMR domains act as conserved ribosome collisions sensors in bacteria, not all have nuclease activity. We do not see any evidence that MutS2 targets mRNAs encoding stalling sequences for degradation as SmrB does so effectively in *E. coli*. Consistent with this, although the DxH residues associated with SmrB endonuclease activity in proteobacteria are also conserved in Bacteroidetes proteins with the KOW-SMR architecture, they are only rarely present in the proteins with the MutS2 architecture. Substitution of the histidine in the DxH motif required for metal-independent catalysis appears to be a repeated theme throughout the SMR family, occurring several times in various lineages. The SmrA proteins in gammaproteobacteria, for example, which are paralogs of SmrB, wholly lack the residues necessary for nuclease activity. In plants, SMR domains display a diversity of active sites,

with some retaining the DxH, others containing the same DxR motif reported here for MutS2, and still others with further substitutions of these residues (Glover et al, 2020).

Based on the growing evidence for the role of SMR domains in sensing ribosome collisions, we propose that SMR domain proteins participate in at least two pathways. The active versions, like SmrB, Cue2, and Nonu-1, work via mRNA cleavage at collisions. In bacteria, cleavage leads to ribosome rescue by the tmRNA pathway; in eukaryotes, the active SMR versions likely function along with the exosomal mRNA degradation system conserved in the archaeo-eukaryotic lineage. In contrast, the inactive versions, like MutS2, are likely to depend on ribosome-splitting pathways coupled with the ancient RqcH/Rqc2 pathway that was present in the last universal common ancestor. While MutS2 carries its own ABC ATPase domain, critical for ribosome splitting, in eukaryotes the inactive SMR domains could function along with related but distinct ribosome-splitting enzymes of the translation factor ABC ATPase clade (e.g., Rli1 in yeast and ABCE1 in humans). Thus, the SMR domains parallel the evolution of the RNase H-fold release factor (eRF1) family (Burroughs and Aravind, 2019), which also features both catalytically active versions involved in the release of the polypeptide from the tRNA (e.g., eRF1) and inactive versions that separate ribosomal subunits (e.g., Dom34 in yeast and PELO in humans).

# Methods

## Reagents and tools

See Table 1

## Table 1. Reagents and tools.

| Reagent/resource | Reference or source | Identifier or catalog number |
|---|---|---|
| **Experimental models** | | |
| *B. subtilis* WT 168 | Bacillus Genetic Stock Center | 1A1 |
| *B. subtilis* ΔmutSB::kan | Bacillus Genetic Stock Center | BKK28580 |
| *B. subtilis* ΔyloA::kan | Bacillus Genetic Stock Center | BKK15640 |
| *B. subtilis* ΔsmpB::kan | Bacillus Genetic Stock Center | BKK33600 |
| *B. subtilis* ΔmutSB::kan thrC::FLAG-MutS2 | This study | FLAG-MutS2 |
| *B. subtilis* ΔmutSB::kan thrC::FLAG-MutS2(KOW-mut) | This study | KOW-mut |
| *B. subtilis* ΔmutSB::kan thrC::FLAG-MutS2(ΔSMR) | This study | ΔSMR |
| *B. subtilis* ΔmutSB::kan thrC::FLAG-MutS2(ALA) | This study | ALA |
| *B. subtilis* ΔmutSB::kan thrC::FLAG-MutS2(H743A) | This study | H743A |
| *B. subtilis* ΔmutSB::kan thrC::FLAG-MutS2(Walker B) | This study | Walker B |
| *B. subtilis* WT 168 amyE::Pveg-NanoLucBleR | This study | WT NonStall |
| *B. subtilis* WT 168 amyE::Pveg-NanoLuc-STOP-BleR | This study | WT STOP |
| *B. subtilis* WT 168 amyE::Pveg-NanoLuc-ApdA-BleR | This study | WT ApdA |
| *B. subtilis* ΔmutSB::kan amyE::Pveg-NanoLucBleR | This study | ΔmutS2 Nonstall |

**Table 1.** (continued)

| Reagent/resource | Reference or source | Identifier or catalog number |
|---|---|---|
| *B. subtilis* ΔmutSB::kan amyE::Pveg-NanoLuc-STOP-BleR | This study | ΔmutS2 STOP |
| *B. subtilis* ΔmutSB::kan amyE::Pveg-NanoLuc-ApdA-BleR | This study | ΔmutS2 ApdA |
| *B. subtilis* ΔmutSB::kan thrC::FLAG-MutS2 amyE:: Pveg-NanoLuc-ApdA-BleR | This study | |
| *B. subtilis* ΔmutSB::kan thrC::FLAG-MutS2(Walker B) amyE:: Pveg-NanoLuc-ApdA-BleR | This study | |
| *B. subtilis* ΔmutSB::kan thrC::FLAG-MutS2(KOW-mut) amyE::Pveg-NanoLuc-ApdA-BleR | This study | |
| *B. subtilis* ΔmutSB::kan thrC::FLAG-MutS2(ΔSMR) amyE::Pveg-NanoLuc-ApdA-BleR | This study | |
| *B. subtilis* ΔmutSB::kan thrC::FLAG-MutS2(ALA) amyE::Pveg-NanoLuc-ApdA-BleR | This study | |
| *B. subtilis* ΔmutSB::kan thrC::FLAG-MutS2(H743A) amyE::Pveg-NanoLuc-ApdA-BleR | This study | |
| **Recombinant DNA** | | |
| pDG1662 | Bacillus Genetic Stock Center | ECE113 |
| pDG1731 | Bacillus Genetic Stock Center | ECE119 |
| pNonstall | This study | |
| pSTOP | This study | |
| pApdA | This study | |
| pFLAG-MutS2 | This study | |
| pFLAG-MutS2(Walker B) | This study | |
| pFLAG-MutS2(KOW-mut) | This study | |
| pFLAG-MutS2(ΔSMR) | This study | |
| pFLAG-MutS2(ALA) | This study | |
| pFLAG-MutS2(H743A) | This study | |
| pET24d(+) His-MutS2 | This study | |
| pET24d(+) His-MutS2(ALA) | This study | |
| pET24d(+) His-MutS2(Walker A) | This study | |
| pET24d(+) His-MutS2(Walker B) | This study | |
| **Antibodies** | | |
| Anti-FLAG HRP-conjugated | MilliporeSigma | A8592 |
| Anti-NanoLuc | Promega | N7000 |
| Anti-FtsZ | MilliporeSigma | ABS2200 |
| Anti-rabbit IgG HRP-conjugated (secondary) | Santa Cruz Biotechnology | sc-2357 |
| Anti-mouse IgG HRP-conjugated (secondary) | Thermo Fisher | 32430 |
| **Oligonucleotides and other sequence-based reagents** | | |
| pDG1731 FLAG-MutS2 primer 1 | This study | cttcggatcctagaagcttatcgAACCGCTGTACGTCATTTTC |
| pDG1731 FLAG-MutS2 primer 2 | This study | GTCATCGTCATCTTTATAATCcatCGTTTCTCCTCCATTCCGCG |
| pDG1731 FLAG-MutS2 primer 3 | This study | CGCGGAATGGAGGAGAAACGatgGATTATAAAGATGACGATGACAAGC |
| pDG1731 FLAG-MutS2 primer 4 | This study | gccagggctgcaggaattCAAAAAAAAGAAAAAGAAAGGTAAATCTTTCTTTTTTttaTTTTAGTTCAACAACCGTAACGCC |
| pDG1662 Nonstall reporter primer 1 | This study | GatcctagaagcttatcgaattttgtcaaaataatttattgacaacgtcttattaacgttgataccggttaaattttatttgacaaaaatgggctcgtgttgtacaataaatgtGATTAACTAATAAGGAGGACAAACATGAGATCTagcgcttggag |

**Table 1.** (continued)

| Reagent/resource | Reference or source | Identifier or catalog number |
|---|---|---|
| pDG1662 Nonstall reporter primer 2 | This study | gataagctgtcaaacatgagaattAAAAAAAAAGCCCGC TCATTAGGCGGGCTtttgtcatcGtcAtcTttAtaAtc GATACAAATCCTGATttatttgtcatcGtcAtcTttAtaAtcG |
| ApdA sequence | This study | TCAGAACTGCACGACGAAGCGGCGCCGGCGTCAAGA ACAGCGAACAGACTGGCGCTGGGAGACAGAGCGCCGCC GTTTCCGGTGGCGGTG |
| MifM Fwd primer | This study | GAAATTAATACGACTCACTATAGGG |
| MifM Rev primer | This study | TTATTATTATTAGTCTTCCTCATCG |
| **Chemicals, enzymes, and other reagents** (*e.g., drugs, peptides, recombinant proteins, dyes etc.*) | | |
| RNase A | Thermo Fisher | EN0531 |
| SuperaseIN | Thermo Fisher | AM2696 |
| Bortezomib | MilliporeSigma | 5.04314 |
| CDP-Star | MilliporeSigma | 12041677001 |
| Lysozyme | MilliporeSigma | L6876 |
| **Software** *Include version where applicable* | | |
| EPU | Thermo Fisher | |
| MotionCor2 | https://emcore.ucsf.edu/ucsf-software Zheng et al (2017) | |
| Gctf | https://www2.mrc-lmb.cam.ac.uk/ download/gctf/ Zhang (2016) | |
| Gautomatch | http://www.mrclmb.cam.ac.uk/ kzhang/ | |
| RELION | https://github.com/3dem/relion Zivanov et al (2018) | |
| CryoSPARC | https://cryosparc.com/ Punjani et al (2017) | |
| ChimeraX | https://www.rbvi.ucsf.edu/chimerax/ Goddard et al (2018) | |
| PHENIX 1.20.1 | http://www.phenix-online.org/ Adams et al (2010) | |

## Bacterial strains and plasmids

A list of strains and plasmids and the details of their construction are given in the Reagents and Tools Table (Table 1). Knockout strains were obtained from the Bacillus Genetic Stock Center (BGSC) (Koo et al, 2017). The reporter constructs and CamR marker were introduced into the *amyE* locus through recombination (Guérout-Fleury et al, 1996) and verified by PCR and Sanger sequencing. All reporter constructs were expressed from a $P_{veg}$ promoter and a strong ribosome binding site (RBS) (Guiziou et al, 2016). N-terminal Flag-tagged versions of MutS2 with a spectinomycin resistance marker were introduced into Δ*mutS2* cells into the *thrC* locus with the endogenous *mutS2* 5' UTR and terminator by recombination (Guérout-Fleury et al, 1996) and confirmed by PCR and Sanger sequencing.

## Spotting assays

Cells were grown overnight at 37 °C in liquid LB. The overnight cultures were diluted to prepare fivefold serial dilutions starting from $OD_{600} = 0.005$. Subsequently, 1.5 μl of the diluted cultures was spotted on LB plates with or without various antibiotics. Plates were then incubated at 37 °C overnight.

## Polysome profiling

*B. subtilis* cells were cultured at 37 °C in 500 mL of LB to $OD_{600} = 0.45$, at which point the cells were treated for 5 min with antibiotics at the concentrations indicated in the figures. Cells were harvested by filtration using a Kontes 99-mm filtration apparatus with a 0.45-μm nitrocellulose filter (Whatman) and flash-frozen in liquid nitrogen. Cells were then lysed in lysis buffer (20 mM Tris pH 8.0, 10 mM $MgCl_2$, 100 mM $NH_4Cl$, 5 mM $CaCl_2$, 0.4% Triton X-100, 0.1% NP-40, 1 mM CAM, 100 U $ml^{-1}$ DNase I) using a Spex 6875D freezer mill with six cycles of 1 min grinding at 6 Hz and 1 min cooling. Lysates were centrifuged at 20,000×*g* for 10 min at 4 °C to pellet cell debris. Samples that were subjected to RNase A digestion to detect nuclease-resistant disomes were incubated for 1 h at 25 °C with 15 μl of RNase A (1:1000 dilution) then treated with 6 μl of SUPERaseIn RNase Inhibitor (Thermo Fisher). Sucrose gradients of 10–50% were prepared using a Gradient Master 108 (Biocomp) with gradient buffer (20 mM Tris pH 8.0, 10 mM $MgCl_2$, 100 mM $NH_4Cl$, 2 mM DTT). Then, 15–30 AU of lysate was loaded on top of the sucrose gradient and centrifuged in an SW 41 rotor at 35,000 rpm for 2.5 h at 4 °C. Fractionation was performed on a Piston Gradients Fractionator (Biocomp). To process each fraction for western blots, proteins

were precipitated in 10% TCA. After pelleting, pellets were washed twice in ice-cold acetone and vacuum-dried for 5 min. Finally, we resuspended each pellet in 6× loading dye and neutralized the solution with Tris-HCl pH 7.5. Samples were probed on western blots using an anti-Flag-HRP antibody (1:2000 dilution) and detected using SuperSignal West Femto Maximum Sensitivity Substrate (Thermo Fisher) and visualized using the ChemiDoc Imaging System (Biorad).

## Western blots

Cells were grown in LB with appropriate antibiotics to $OD_{600}$ ~1. In total, 1 mL of culture was harvested by centrifugation, resuspended in lysis buffer (100 mM NaCl, 50 mM EDTA) with 7 µl of 10 mg ml$^{-1}$ lysozyme and incubated at 37 °C for 30 min. After 40 µl of 20% sarkosyl was added, the samples were incubated for 5 min at 37 °C. Then, 6× loading dye (250 mM Tris pH 6.8, 20% glycerol, 30% β-mercaptoethanol, 10% SDS, saturated bromophenol blue) was added and the lysate was denatured at 90 °C for 10 min. Protein was separated on either a 4–12% or 12% Criterion XT Bis-Tris protein gel (Bio-Rad) with XT-MES buffer and transferred to polyvinylidene membranes using the Trans-Blot Turbo Transfer system (Bio-Rad). Membranes were blocked in 5% milk for 1 h at room temperature, washed, and then probed with antibodies diluted in TBS-Tween at the following dilutions: anti-NanoLuc, 1:2,000 (Promega); anti-FtsZ, 1:2000 (Sigma); anti-mouse-HRP, 1:4000 (Thermo Fisher); anti-rabbit-HRP, 1:4000 (Santa Cruz Biotechnologies). Chemiluminescent signals from HRP were detected using SuperSignal West Pico PLUS Chemiluminescent Substrate (Thermo Fisher) or SuperSignal West Femto Maximum Sensitivity Substrate (Thermo Fisher) and were visualized using the ChemiDoc Imaging System (Bio-Rad).

## Northern blots

Cells were grown in LB to $OD_{600}$ = 0.5, an equal volume of ice-cold methanol was added, and the samples were harvested by centrifugation. Pellets were frozen on dry ice and stored at −80 °C. Pellets were then thawed on ice and resuspended in lysis buffer (30 mM Tris pH 8.0, 10 mM EDTA). An equal amount of lysis buffer with 10 mg ml$^{-1}$ lysozyme was added to the lysates and incubated at 37 °C shaking at 1000 rpm for 30 min. RNA was extracted twice with phenol (pH 4.5) first at 65 °C and then at room temperature, followed by chloroform extraction. The RNA in the aqueous layer was precipitated with isopropanol and 0.3 M sodium acetate (pH 5.5), washed with 80% ethanol, and resuspended in water. The purified RNA was separated on a 1.2% agarose-formaldehyde denaturing gel and was then transferred to a nylon membrane (Hybond-N +, Cytvia) in 10× SSC buffer using a model 785 vacuum blotter (Bio-Rad). RNA was cross-linked to the membrane using an ultraviolet (UV) cross-linker (Stratgen). Pre-hybridization and hybridization were performed in PerfectHyb Plus Hybridization Buffer (Sigma). The RNA was probed with 50–150 nM 5'-digoxigenin-labeled DNA oligonucleotides (IDT). Digoxigenin was detected with anti-digoxigenin-AP antibodies diluted 1:500–1:1000 (Sigma). Chemiluminescent signals from alkaline phosphatase were detected with CDP-star (Sigma) and were visualized using the ChemiDoc Imaging System (Bio-Rad).

## MS analysis of tagging sites on the reporter protein

Strains expressing the ApdA reporter were grown in 100 mL of LB with 20 µM bortezomib until $OD_{600}$ = 0.5 and harvested by centrifugation. The pellet was frozen at −80 °C and thawed in 2× CellLytic B-cell lysis reagent (Sigma) and 0.2 mg mL$^{-1}$ lysozyme for 10 min. The lysate was clarified by centrifugation for 30 min at 20,000×$g$ at 4 °C. In total, 50 µL of Strep-tactin Sepharose beads (IBA) were added to the supernatant and samples were incubated at 4 C for 1 h. The beads were washed four times with IP wash buffer (20 mM Tris pH 8.0, 100 mM $NH_4Cl$, 0.4% Triton X-100, 0.1% NP-40) for 5 min at 4 °C. Protein was eluted from the beads by shaking at 4 °C in elution buffer (20 mM Tris pH 8.0, 100 mM $NH_4Cl$, 5 mM desthiobiotin) for 1 h. Then, 36 µl of the immunoprecipitated sample was reduced with 100 mM DTT in 100 mM triethylammonium bicarbonate (TEAB) buffer at 58 °C for 55 min and then the pH was adjusted to 8.0. The samples were alkylated with 200 mM iodoacetamide in 100 mM TEAB buffer in the dark at room temperature for 15 min. Proteins were pelleted and resuspended in 50 mM TEAB and proteolyzed with 15 ng µL$^{-1}$ of LysC (Wyco) at 37 °C overnight. Peptides were desalted on Oasis u-HLB plates (Waters), eluted with 60% acetonitrile (ACN)/0.1% trifluoroacetic acid (TFA), dried and reconstituted in 2% ACN/0.1% formic acid.

### LC-MS/MS analysis

Desalted peptides cleaved by LysC were analyzed by LC-MS/MS. Then peptides were separated by reverse-phase chromatography (2–90% ACN/0.1% formic acid gradient over 63 min at a rate of 300 nL min$^{-1}$) on a 75 µm × 150 mm ReproSIL-Pur-120-C18-AQ column (Dr. Albin Maisch, Germany) using the nano-EasyLC 1200 system (Thermo). Eluting peptides were sprayed into an Orbitrap-Lumos_ETD mass spectrometer through a 1-µm emitter tip (New Objective) at 2.4 kV. Scans were acquired within 350–1600 Da $m/z$ targeting the truncated reporter with 15 s dynamic exclusion. Precursor ions were individually isolated and were fragmented (MS/MS) using an HCD activation collision energy of 30. Precursor (fragment) ions were analyzed at a resolution of 200 Da of 120,000 with the following parameters: max injection time (IT), 100 ms (resolution of 30,000) in three cycles. The MS/MS spectra were processed with Proteome Discover v2.4 (Thermo Fisher) and were analyzed with Mascot v.2.8.0 (Matrix Science) using RefSeq2021_204_Bacillus.S and a database with peptides from the NanoLucBleR reporter protein. Peptide identifications from Mascot searches were processed within the Proteome Discoverer-Percolator to identify peptides with a confidence threshold of a 5% false discovery rate, as determined by an auto-concatenated decoy database search.

## Purification of MutS2

N-terminally His-tagged versions of MutS2 were expressed from pET24d(+) plasmids in BL21(DE3) *E. coli* cells. Cells were grown in 9 L LB medium to approximately $OD_{600}$ = 2.5 and MutS2 expression was induced with IPTG (1 mM). Cells were harvested by centrifugation and resuspended in 30 mL lysis buffer (20 mM Hepes pH 7.5, 95 mM KCl, 5 mM $NH_4Cl$, 10 mM Mg(OAc)$_2$, 1 mM DTT, protease inhibitor (Roche)), then lysed using a microfluidizer (15k psi, H10Z, Microfluidics). Lysates were cleared by centrifugation (16,000 rpm, 20 min, 4 °C, Sorvall SS-34 rotor). Cleared lysates

were applied to 3 mL TALON Metal Affinity Resin (Takara) and incubated for 30 min at 4 °C. The resin was washed with 40 mL each of wash buffer (50 mM Hepes pH 7.5, 225 mM NH$_4$Cl, 20 mM MgCl$_2$, 0.1 mM PMSF, 5% glycerol, 1 mM DTT, 20 mM imidazole, 0.4% Triton X-100, protease inhibitor), wash buffer with 1 M KCl, and wash buffer without imidazole or Triton X-100, sequentially. The protein was eluted by incubation with 5 mL wash buffer with 150 mM imidazole, followed by a second elution with 200 mM imidazole. Elution fractions were analyzed by Superdex 200 gel filtration and fractions containing pure MutS2 protein were pooled, concentrated using an Amicon 50 kDa MWCO concentrator, and used for the cryo-EM and subunit splitting experiments.

### *B. subtilis* translation extract preparation

*Bacillus subtilis* strain 168 Δ*hpf* Δ*ssrA* Δ*SAS1-2* cells were grown on an LB agar plate containing 5 µg/mL CAM, 1 µg/mL erythromycin and 10 µg/mL kanamycin at 37 °C. Six 2 L flasks with LB medium supplemented with 1% glucose were inoculated to an OD$_{600}$ of 0.02 and incubated at 37 °C. Cells were harvested during the exponential growth phase at an OD$_{600}$ between 1 and 2 by centrifugation at 8000 rpm for 5 min at room temperature. Cells were resuspended in 1× PBS, combined into one bottle, and pelleted again. Cells were resuspended in lysis buffer (10 mM HEPES-KOH pH 8.2, 60 mM K-glutamate, 14 mM Mg(OAc)$_2$ and 20 µg/ml DNase I). Cells were lysed by one pass through a cell disruptor (Constant Systems) at 18,000 psi at room temperature, with the lysate collected on ice. Lysate was centrifuged in an SS-34 rotor at 18,000 rpm for 15 min at 4 °C. Extracts were frozen in liquid nitrogen and stored at −80 °C.

### Preparation of mRNA construct

mRNA was prepared by PCR amplification of DNA template followed by in vitro transcription and precipitation of mRNA using LiCl. For *B. subtilis* disomes, an mRNA was transcribed from pMAT MifM_WT_V5 encoding a His-tag, 3C cleavage site, V5 tag for detection, and MifM stalling sequence. The forward primer was MifM_for2, annealing to the T7 promoter. For short/standard length mRNA used in splitting assays, a reverse primer was used to append four stop codons immediately downstream of the stalling sequence (MifM_rev2). In vitro transcription reactions were set up in 100 µL reactions with 2 µg DNA, 3 µg T7 RNA Polymerase and performed in 40 mM Tris pH 7.9, 2.5 mM Spermidine, 26 mM MgCl$_2$, 0.01% Trition X-100, 5 mM DTT, 8 mM each ATP, GTP, CTP, UTP, and 0.4 U/µL SUPERaseIn RNase Inhibitor (Invitrogen). Reactions were incubated for 4 h at 37 °C. RNA was precipitated by the addition of 150 µL 7 M LiCl and incubated at −20 °C overnight, then washed with 70% ethanol and resuspended in water.

### *B. subtilis* in vitro translation and isolation of disomes

Diomes were generated by performing in vitro translation with an mRNA featuring a MifM stalling sequence and His-tag for purification of programmed ribosomes via the nascent chain. In vitro translation reactions were prepared at room temperature in a final volume of 12 mL, with 4700 A260 units cell extract, 480 µg mRNA, 50 mM HEPES pH 8.2, 10 mM NH$_4$OAc, 130 mM

potassium acetate, 30 mM sodium pyruvate, 4 mM sodium oxalate, 50 µg/mL tRNA (from *E. coli* MRE 600 – Sigma 10109541001), 0.2 mg/mL folinic acid, 0.1 mg/mL creatine kinase, 20 mM creatine phosphate, 4 mM ATP, 3 mM GTP, 0.15 mM amino acids, 1 mM DTT, 0.08 U/mL SUPERaseIn RNase Inhibitor (Invitrogen), 1% PEG 8000 and magnesium acetate (typically 16–20 mM in addition to that present in extract, determined by performing test translations monitored by Western blot). The reaction was split to 1 mL aliquots, and incubated at 32 °C for 40 min, shaking at 1000 rpm. In vitro translation reaction was incubated at 4 °C for 1 h with TALON resin (3.8 mL slurry) that had been washed with buffer A (30 mM HEPES pH 7.5, 100 mM KOAc, 10 mM Mg(OAc)$_2$, 0.2% DDM) and, incubated with *E. coli* tRNA to reduce non-specific binding of ribosomes. Talon beads were then washed with 30 mL buffer A supplemented with 20 mM imidazole. Elution was performed by cleavage with 3C protease (3.3 mg in 5 mL buffer A) at 4 °C. The sample was loaded on 10–50% sucrose gradients prepared in buffer A and centrifuged overnight in a SW40 rotor at 4 °C (for an equivalent of 3 h at 40,000 rpm). Gradients were fractionated on a Biocomp Gradient Station and the disome peak was collected. Disomes were pelleted in a TLA110 rotor for 1 h at 100,000 rpm at 4 °C and resuspended in 25 mM HEPES pH 7.5, 150 mM KOAc, 10 mM Mg(OAc)$_2$, 2 mM DTT. If not used immediately, disomes were frozen in liquid nitrogen and stored at −80 °C.

### In vitro reconstitution of the MutS2-disome complex

His-MutS2 WT was added to purified disomes from *B. subtilis* in tenfold excess in reaction buffer (50 mM HEPES/KOH pH 7.5, 75 mM KOAc, 5 mM Mg(OAc)$_2$, 1.2 mM DTT, 45 mM NH$_4$Cl, 4 mM MgCl$_2$, 1% glycerol, 1 mM MnCl$_2$, 1 mM AMP-PNP) and the mixture was incubated at 30 °C for 1 h. Following this, the sample was directly vitrified for cryo-EM by plunge-freezing using a Vitrobot Mark IV (FEI Company/Thermo Fisher) with an incubation time of 45 s and blotting for 2.5 s at 4 °C and a humidity of 95%.

### Purification of cross-linked FLAG-MutS2-disome complexes from *Bacillus subtilis* for cryo-EM

*B. subtilis* cells expressing N-terminally FLAG-tagged MutS2 with the E416A mutation in the Walker B motif were grown in 4 L LB medium supplemented with kanamycin for selection to an OD of 1.5. Cells were spun down at 7800×*g* for 10 min at room temperature. Pellets were washed with PBS and pooled, then pelleted again and frozen in liquid nitrogen. Frozen pellets were thawed and resuspended in lysis buffer (20 mM HEPES/KOH pH 7.5, 100 mM KCl, 15 mM Mg(OAc)$_2$, 1 mM DTT, 2 mM spermidine, protease inhibitor cocktail). Cell lysis was performed in a cell disruptor (Constant Systems) at 20,000 psi at room temperature. The following steps were performed at 4 °C. Lysates were cleared by centrifugation at 27,000×*g* for 15 min. Cleared lysates were incubated with 250 µL Anti-FLAG M2 Affinity Gel (Sigma) equilibrated in lysis buffer for 2 h on a wheel. Beads were recovered by centrifugation at 1380×*g* for 3 min and washed with 10 mL lysis buffer as a batch, then transferred to a G-25 column (MoBiCol). The column was washed twice with 10 mL lysis buffer. Elution by incubating with 50 µL FLAG peptide (145 µg/mL) in

lysis buffer for 1 h. Samples were recovered by centrifugation of the column at 376×$g$ for 2 min. Crosslinking was performed by adding 0.5 mM BS3 to the sample and shaking lightly at 10 °C for 30 min. The reaction was quenched by the addition of 25 mM Tris/HCl (pH 7.5) before vitrification. Vitrification of the sample was performed as described for the reconstituted samples.

## Cryo-EM analyses of in vitro reconstituted sample

### Data collection
Data were collected on a Titan Krios G3 (Thermo Fisher) equipped with a K2 direct detector (Gatan) at 300 keV using the semi-automated data acquisition software EPU (Thermo Fisher). 40 frames with a dose of 1.09 $e^-/Å^2$ per frame were collected in a defocus range of −0.4 to −3.5 μm. Magnification settings resulted in a pixel size of 1.045 Å/pixel. Frame alignment was executed with MotionCor2 (Zheng et al, 2017) and the estimation of the contrast transfer function (CTF) was performed with Gctf (Zhang, 2016).

### Processing
After the manual screening of micrographs, 5784 were selected for particle picking using Gautomatch (http://www.mrclmb.cam.ac.uk/kzhang/) with a set of reference images generated from a *B. subtilis* disome model from Saito et al. After 2D classification in Relion 3.1 (Zivanov et al, 2018), 96,678 particles representing collided disomes were selected for further processing. After several rounds of 3D classification and refinement in Relion in order to remove classes without a rigid disome interface or density for the mRNA in the interribosomal space, a class of 11,749 particles displaying a significant extra density next to the interribosomal mRNA were selected for high-resolution refinement in CryoSPARC (Punjani et al, 2017). Homogenous refinement and focused refinement yielded a final reconstruction of the collided disome bound by the MutS2 KOW and SMR domains at an overall resolution of 3.8 Å.

### MutS2 model building
In order to verify the identification of the extra densities observed in the reconstruction of the *B. subtilis* disome as MutS2, structures of the SMR and KOW domains of MutS2 as predicted by Alphafold2 (Jumper et al, 2021) were fitted as rigid bodies into the densities in UCSF ChimeraX (Goddard et al, 2018). A model of the collided disome bound by MutS2 was generated by adding these models to the model of a collided *B. subtilis* disome (Saito et al, 2022) and refining the resulting model in Phenix 1.20.1 (Adams et al, 2010) after minor adjustments using WinCoot 0.8.9.2 and ISOLDE for ChimeraX, based on the experimental data.

## Cryo-EM analyses of cross-linked native sample

Data were collected on a Titan Krios TEM equipped with a Falcon II DED at 300 kV. Ten frames with a dose of 2.5 $e^-/Å^2$ per frame were collected in a defocus range of −0.5 to −4.0 μm. Magnification settings resulted in a pixel size of 1.09 Å/pixel. Frame alignment was executed with MotionCor2 (Zheng et al, 2017) and the estimation of the contrast transfer function (CTF) was performed with Gctf (Zhang, 2016).

After manual screening of micrographs, 11,923 were selected for particle picking using Gautomatch with a set of reference images generated from a *B. subtilis* disome model (Saito et al, 2022). After

2D classification in Relion 3.1 (Zivanov et al, 2018), 82,276 particles representing collided disomes were selected for further processing. 3D classification and refinement in Relion resulted in a class of 19,230 particles displaying the density previously identified as corresponding to MutS2 SMR and KOW domains. This class was used for high-resolution refinement in CryoSPARC (Punjani et al, 2017), resulting in a reconstruction with an overall resolution of 6.8 Å.

## In vitro splitting assays

Purified versions (WT, SMR domain mutants, Walker A and Walker B mutants) of the MutS2 protein were added to purified disomes from *B. subtilis* in tenfold excess in reaction buffer (50 mM HEPES/KOH pH 7.5, 75 mM KOAc, 5 mM Mg(OAc)$_2$, 1.2 mM DTT, 45 mM NH$_4$Cl, 4 mM MgCl$_2$, 1% glycerol, 1 mM MnCl$_2$) together with 1 mM ATP, 1 mM AMP-PNP, or no additional nucleotides and the mixture was incubated at 30 °C for 1 h. Samples were then applied to 10–50% continuous sucrose density gradients (50 mM HEPES/KOH pH 7.5, 100 mM KOAc, 5 mM Mg(OAc)$_2$, 1 mM DTT). The gradients were centrifuged in an SW40Ti rotor (Beckman Coulter) at 202,408×$g$ for 150 min and fractionated using a BioComp Gradient Station while UV absorption at 260 nm was recorded using a Triax Flow Cell FC-2.

# Data availability

Cryo-EM volumes and molecular models have been deposited at the Electron Microscopy Data Bank and Protein Data Bank with accession codes EMD-18585 (reconstituted complex), EMD-18566 (native cross-linked complex), EMD-18558 and PDB-8QPP (stalled 70 S), and EMD-18901 and PDB-8R55 (collided 70 S).

# Peer review information

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

## Acknowledgements

The authors thank Bob Cole and Tatiana Boronina at JHMI in the Mass Spectrometry and Proteomics Facility and Joanna Musial, as well as Hannah Buschkämper for excellent technical assistance. This work was supported by NIH grant GM136960 (ARB), HHMI (RG), the German Research Council (TRR174) (RB), and by the Intramural Research Program of the National Library of Medicine at the NIH (AMB and LA).

## Author contributions

**Esther N Park**: Conceptualization; Resources; Formal analysis; Investigation; Methodology; Writing—original draft; Writing—review and editing. **Timur Mackens-Kiani**: Conceptualization; Resources; Data curation; Formal analysis; Investigation; Visualization; Methodology; Writing—original draft; Writing—review and editing. **Rebekah Berhane**: Investigation; Methodology. **Hanna Esser**: Investigation; Methodology. **Chimeg Erdenebat**: Investigation; Methodology. **A Maxwell Burroughs**: Conceptualization; Formal analysis; Investigation; Writing—original draft. **Otto Berninghausen**: Methodology. **L Aravind**: Conceptualization; Funding acquisition; Investigation; Methodology; Writing—original draft. **Roland Beckmann**: Conceptualization; Supervision; Funding acquisition; Writing—original draft; Writing—review and editing. **Rachel Green**: Conceptualization; Supervision; Funding acquisition; Writing—review and editing. **Allen R Buskirk**: Conceptualization; Supervision; Funding acquisition; Writing—review and editing.

## Disclosure and competing interests statement

The authors declare no competing interests.

# Expanded View Figures

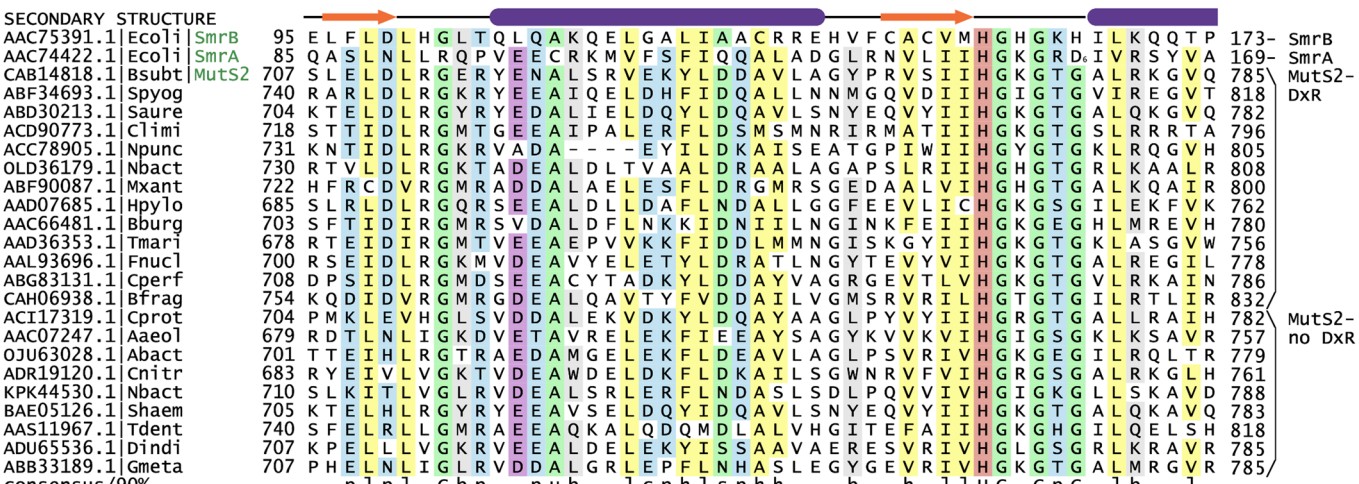

**Figure EV1. Multiple alignment of the conserved residues in the SMR domains from different bacteria.**

Columns in the alignment are shaded and labeled according to biochemical character: —, negatively charged in purple; c, charged in blue; h, hydrophobic in yellow; p, polar in blue; l, aliphatic in yellow; b, big in gray; s, small in green; u, tiny in green; G, glycine in green; H, histidine in red. Sequences are labeled with NCBI accession number and organism abbreviation. Secondary structure provided at top of alignment. Numbers to left and right of alignment denote positioning of the region.

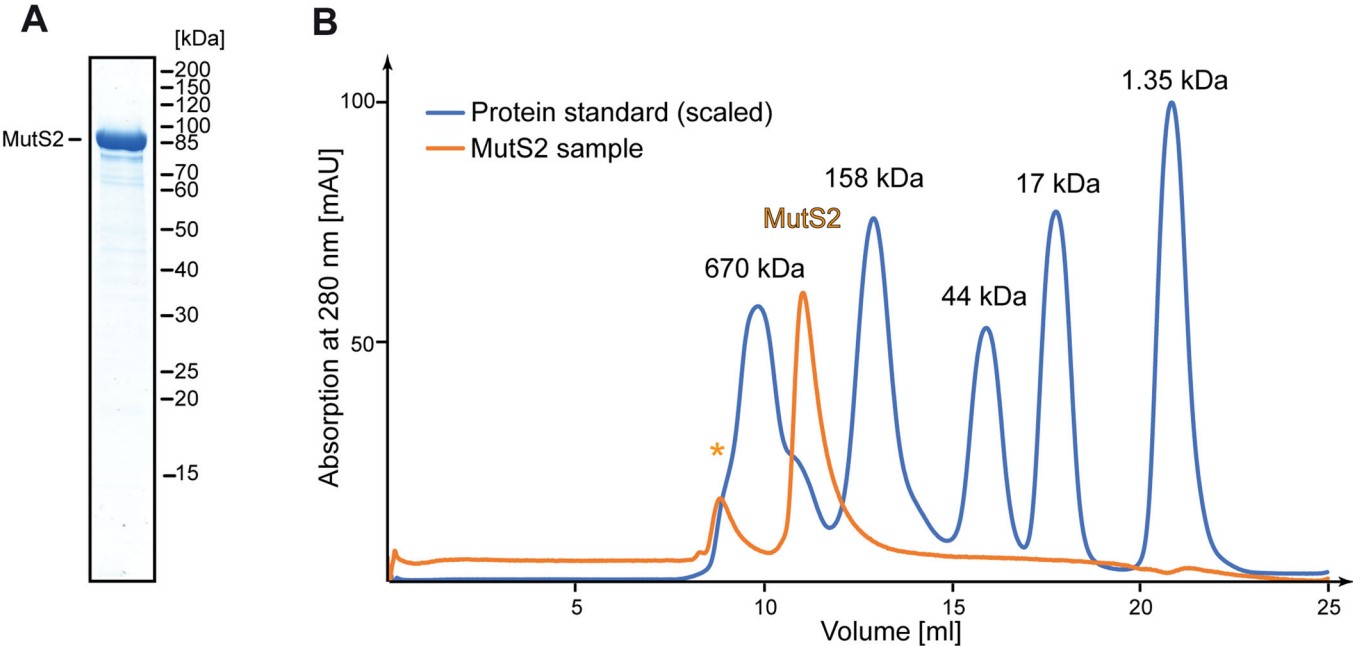

**Figure EV2. MutS2 purification as an oligomer.**

(**A**) Purified sample of *B. subtilis* MutS2 used for reconstitution experiments, shown on a Coomassie-stained 10% Nu-PAGE gel. The expected apparent molecular weight of MutS2 is approximately 87 kDa. (**B**) Chromatograms of size-exclusion chromatography with the purified MutS2 sample (orange) and a protein size standard (blue) consisting of Thyroglobulin (bovine, 670 kDA), g-globulin (bovine, 158 kDa), Ovalbumin (chicken, 44 kDa), Myoglobin (horse, 17 kDa), and Vitamin B12 (1,35 kDa). Samples were analyzed on a Superdex 200 increase 10/300 GL column. Results indicate a single purified MutS2 complex of a size between 158 kDa and 670 kDa, consistent with a dimer or tetramer.

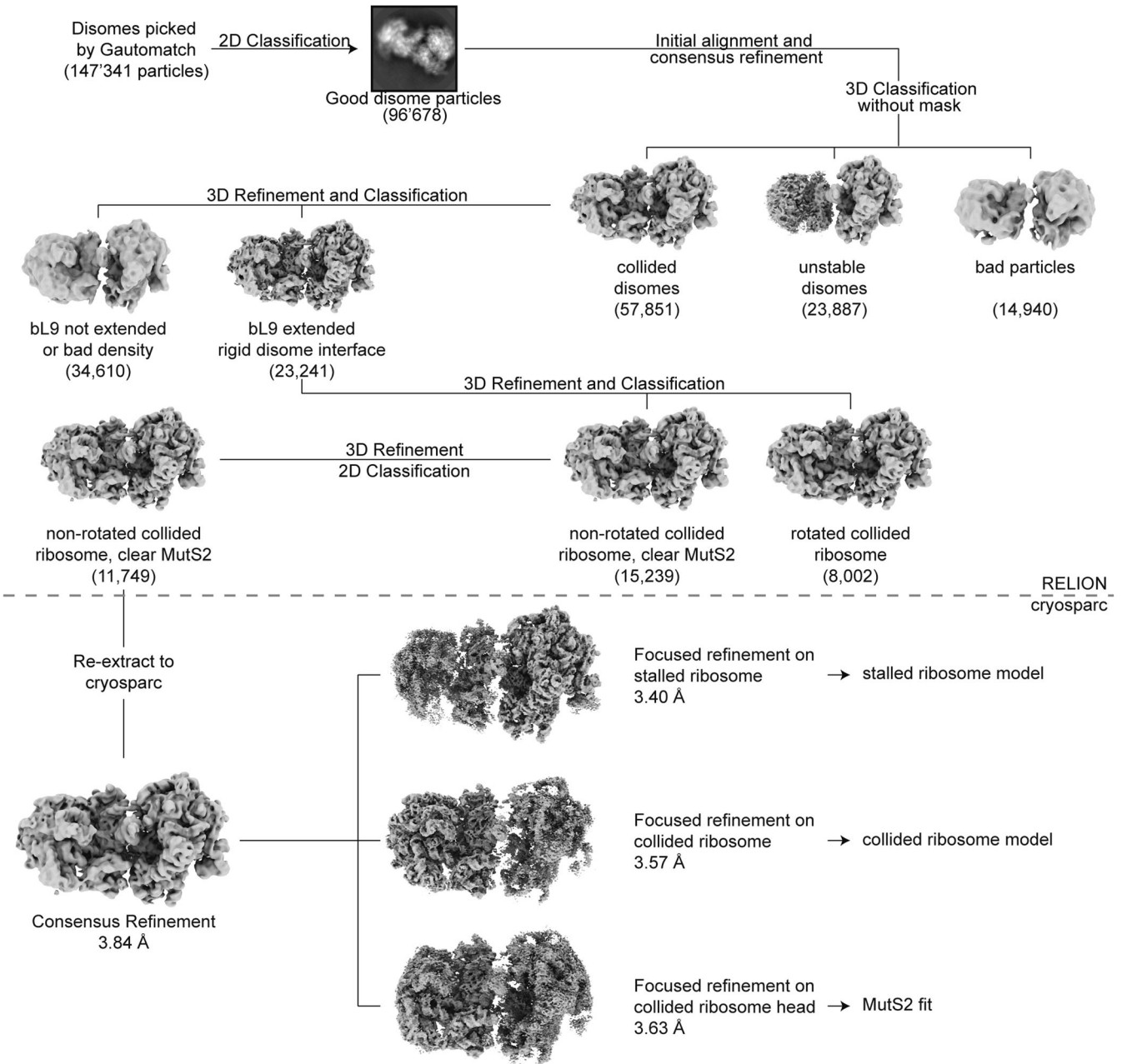

**Figure EV3. Processing scheme for reconstituted MutS2-disome complex.**

Shown are the principal steps of processing as well as representative reconstructions for each step. Initial processing steps and classification were performed in Relion, followed by high-resolution refinement of the resulting class of particles in CryoSPARC.

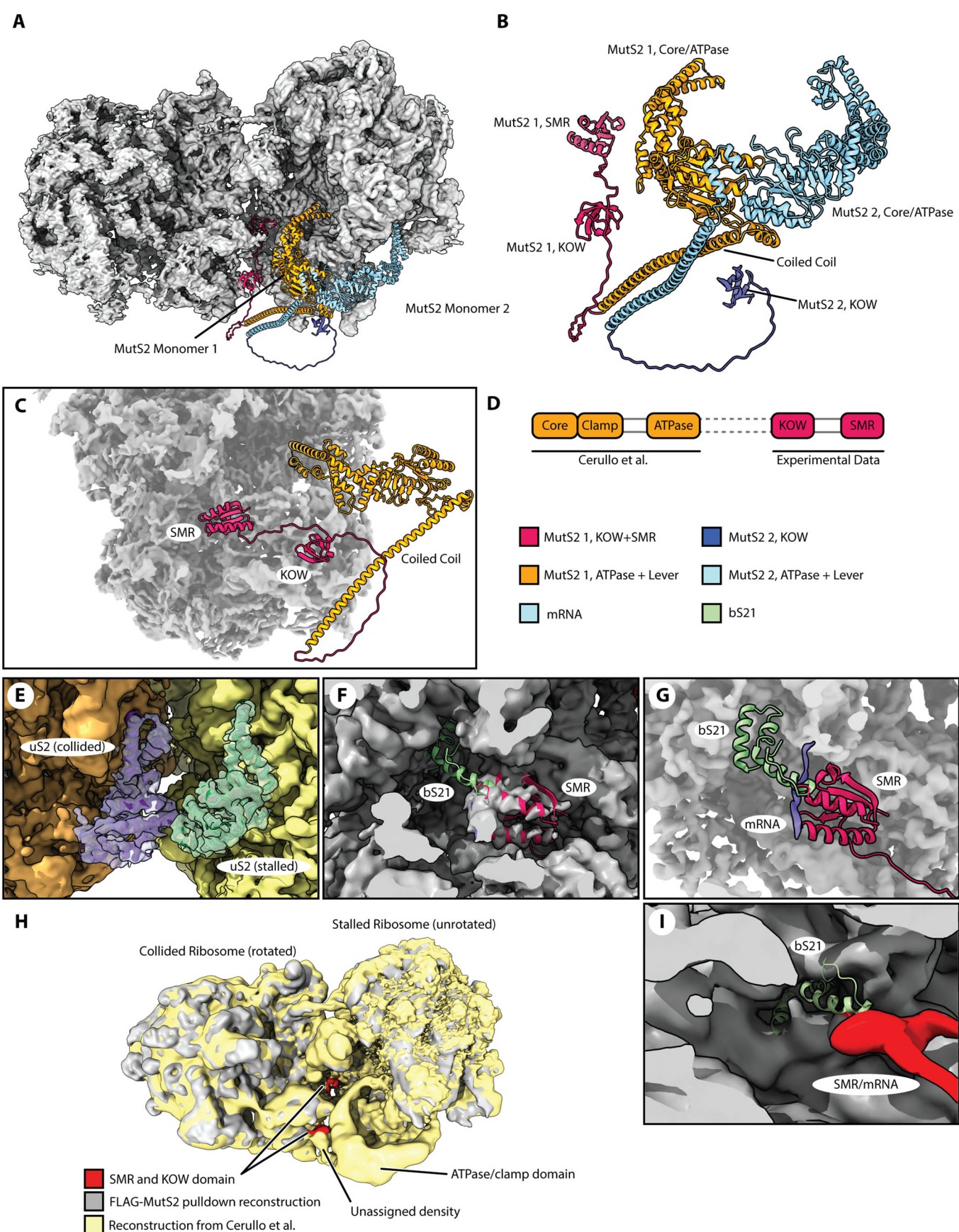

◄ **Figure EV4. MutS2 KOW and SMR domains bind the ribosome in a manner congruent with previous studies on the MutS2 Core/ATPase domains.**

(A) Experimental cryo-EM map (gray) and model of the MutS2 dimer binding a collided disome in *B. subtilis*. The SMR and KOW domains of MutS2 monomer 1 (red) as well as the KOW domain of MutS2 monomer 2 (violet) recruit the MutS2 are visible in the cryo-EM reconstruction. The Core/ATPase domains are not visible in the reconstruction, but the structure as published by Cerullo et al (monomer 1: yellow, monomer 2: light blue) is congruent with our experimental observations. (B) Isolated view of the composite structure of the MutS2 dimer: The length of the flexible loop between coiled-coil and KOW domains does not allow a stringent assignment of either KOW domain to either monomer from the Cerullo et al structure, hence the assignment shown here was chosen arbitrarily. (C) Side view of MutS2 monomer 1 engaged with the stalled ribosome. (D) Schematic representation of the composite structure of MutS2 shown in (A, B). (E) Map-to-model fit of uS2 from both stalled and collided ribosome to a composite map of the MutS2-bound *B. subtilis* disome. (Stalled and collided ribosome maps were refined separately). uS2 is clearly present on both ribosomes in the complex. (F) Fit of the MutS2 monomer 1 SMR domain into the experimental density and comparison with the hypothetical location of bS21 as observed by Cerullo et al. In our experimental data, there is no evidence that bS21 is present in the MutS2-bound collided disomes. (G) Representation of the experimentally determined location of the MutS2 monomer 1 SMR domain and the position of bS21 as observed in Cerullo et al. bS21 would clash with the observed conformation of MutS2 SMR next to the mRNA. (H) Overlay of the experimental cryo-EM map of the MutS2-bound disome published by Cerullo et al and the experimental map of disomes collected from a FLAG-MutS2 pulldown in our experiments. SMR and KOW domain are highlighted in our experimental map. Both ribosomes match in their rotation states, the stalled ribosome being non-rotated in both maps and the collided ribosome rotated. MutS2 ATPase/Core/Clamp domains are visible only in the reconstruction from Cerullo et al, while the SMR domain is visible in our reconstruction. An extra density in the map of Cerullo et al that was not assigned in the original publication is visible close to the position of the KOW domain identified in our data. (I) Close-up view of the binding pocket for bS21 in our in vivo dataset. Only very weak partial density can be observed for bS21 compared to surrounding ribosomal proteins and the density corresponding to mRNA and the MutS2 SMR domain.

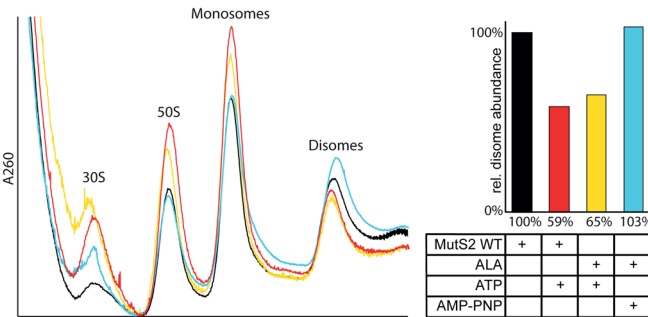

**Figure EV5. The DLR motif of MutS2 SMR domain is not essential for disome splitting.**

Left: UV chromatograms from sucrose gradient fractionation of disome splitting assays with MutS2 WT and MutS2 $D_{711}LR$ to $A_{711}LA$ mutant ("ALA"). Right: Relative abundance of disomes compared to total ribosomal fractions after splitting reaction, calculated from relative peak areas in the chromatograms. Purified *B. subtilis* disomes were used as input. The presence of the mutation has no effect on the efficiency of the splitting reaction either with or without hydrolysable ATP, indicating that the DLR motif of the SMR domain is not required for this process.

