## [Peer Review File · The EMBO Journal]

B. subtilis MutS2 splits stalled ribosomes into subunits without mRNA cleavage

Esther Park, Timur Mackens-Kiani, Rebekah Berhane, Hanna Esser, Chimeg Erdenebat, A. Maxwell Burroughs, Otto Berninghausen, L. Aravind, Roland Beckmann, Rachel Green, and Allen Buskirk

DOI: [10.15252/embj.2023114458](https://doi.org/10.15252/embj.2023114458)

Corresponding author(s): Allen Buskirk (buskirk@jhmi.edu)

Review Timeline:

Submission Date:	6th May 23
Editorial Decision:	4th Jul 23
Revision Received:	3rd Oct 23
Editorial Decision:	7th Nov 23
Revision Received:	16th Nov 23
Accepted:	21st Nov 23

Editor: Cornelius Schneider

Transaction Report:

Dear Dr. Buskirk,

Thank you for submitting your manuscript for consideration by the EMBO Journal.

We apologize for our delay which was caused by a lack of response from a referee which had originally agreed to review the manuscript. In the interest of time, we have decided to proceed with the comments from two reviewers, which are included below for your information.

As you will see from the reports, the reviewers appreciate the work, while also indicating a number of constructive points that would need to be addressed before acceptance here. From my side, I find the reviewer comments reasonable and constructive. Therefore, based on these positive assessments, I would like to invite you to address the issues raised by the reviewers in a revised manuscript. Please note that the missing referee report would have to be taken into account should it still reach us in the near future. I would be happy to discuss the revision in more detail via email or phone/videoconferencing.

We generally allow three months as standard revision time. As a matter of policy, competing manuscripts published during this period will not negatively impact on our assessment of the conceptual advance presented by your study. However, please contact me as soon as possible upon publication of any related work to discuss the appropriate course of action. Should you foresee a problem in meeting this three-month deadline, please contact us to arrange an extension.

When preparing your letter of response to the referees' comments, please bear in mind that this will form part of the Review Process File and will therefore be available online to the community. For more details on our Transparent Editorial Process, please visit our website: <https://www.embopress.org/page/journal/14602075/authorguide#transparentprocess>. Please also see the attached instructions for further guidelines on preparation of the revised manuscript.

Please feel free to contact me if you have any further questions regarding the revision. Thank you for the opportunity to consider your work for publication, and I look forward to your revision.

Yours sincerely,

Cornelius Schneider

Cornelius Schneider, PhD
Editor
The EMBO Journal
c.schneider@embojournal.org

We realize that it is difficult to revise to a specific deadline. In the interest of protecting the conceptual advance provided by the work, we recommend a revision within 3 months (2nd Oct 2023). Please discuss the revision progress ahead of this time with the editor if you require more time to complete the revisions. Use the link below to submit your revision:

Referee #2:

Ribosome quality control mechanisms are essential in all forms of life and include mechanisms to recognize stalled ribosomes via ribosome collisions. The system in *E. coli* responsible for detecting collided ribosomes is not conserved across bacteria, prompting the authors to explore the mechanisms used in other bacteria such as *B. subtilis*. The authors use a combination of bioinformatics, biochemistry, cryo-EM and cell-based experiments to reveal the molecular mechanisms used by the protein MutS2 in *B. subtilis* to detect and clear collided ribosomes stalled on mRNAs. The mechanism of MutS2 differs substantially from that of the *E. coli* protein SmrB. MutS2 seems to act as a dimer and detects stalled ribosomes via its SMR domain, with assistance of its KOW domain. In contrast to *E. coli* SmrB, MutS2 does not cleave the inter-ribosome mRNA, but rather splits the stalled ribosome in an ATP-dependent manner via its ABC ATPase domain into free 30S and peptidyl-tRNA-50S complexes. This is then targeted subsequently by the RQC protein RqcH, which adds poly-Ala to the C-terminus of the aberrant truncated protein to facilitate downstream degradation.

Interestingly, the authors can identify the SMR and KOW domains in their reconstituted samples by cryo-EM, in contrast to Cerullo et al. (ref. 33). By contrast Cerullo et al. see the ATPase domains but not the SMR and KOW domains. There are other differences to note. In Cerullo et al., the stalled ("leading") ribosome lacks ribosomal protein uS2 but recruits protein bS21 which is not a normal component of the *B. subtilis* 30S. By contrast, bS21 is absent from the stalled ribosome in the present work and it is not clear if S2 is missing, as observed in Cerullo et al. (Fig. 2C seems to indicate uS2 is missing or highly flexible and could be viewed at lower resolution.)

Overall, this is a well-written paper, but needs a little more clarity on the differences with the Cerullo et al. work. Do the authors here see uS2 on the stalled ribosome or not? Labels in Fig. 2 would help.

Also, what reasons could explain the difference in bS21 binding? Could the structures represent different steps in the collision and splitting reaction pathway? If not, could the difference be explained by how the samples were prepared? For example, Cerullo et al. purified samples directly from *B. subtilis* cells via a FLAG-tagged MutS2, without using sucrose gradients. In the present work, the authors first purified disomes from *in vitro* translation reactions involving purified *B. subtilis* ribosomes and the remainder coming from the *E. coli* PURE system (refs. 28 and 39) and then added a ten-fold excess of tagged MutS2. It's likely sucrose gradients were used, but it's not clear at what step of the process this would have been done. This should be explicitly included in the methods, as it is a complicated method.

Have the authors tried adding back *B. subtilis* bS21 to the *in vitro* translation reactions, to see if it might be bound where they predict a collision based on the present structure? It could be that the SMR domain in Fig. S2 occupies the binding site simply because there is room to fall into that pocket rather than due to excluding bS21. This should be commented on, at the very least.

Minor: There seems to be some text deleted between the bottom of p. 6 and top of p. 7, possibly hidden behind the figure.

Referee #3:

The study by Park et al. dissects the mechanism of a recently identified bacterial ribosome collision sensor named MutS2 from *B. subtilis*. A recent study of this protein (Cerullo et al. *Nature*, 2022) revealed the structure of the homodimeric ATPase domains

bound to a collided pair of ribosomes, but did not structurally or biochemically characterize the C-terminal KOW domain and SMR domain, whose fold suggests an endonucleolytic function. Now, Park et al. show that the SMR domain does not cleave mRNA but rather along with KOW contributes weakly to sensing disomes.

Using cryo-EM they visualize the KOW and SMR domains on disomes and study the function of these domains using genetic and biochemical methods. They first purified a stalled disome complex from *B. subtilis* lysates and incubated it with purified MutS2 and in the presence of AMP-PNP and studied the complex via cryo-EM. Cryo-EM maps of disomes reveal two KOW and one SMR domains bound at the interface of the collided ribosomes, while the ATPase domains are not resolved in direct contrast to Cerullo et al. Notably the MutS2 SMR domain at the dimeric interface is positioned very differently from SmrB, a ribosome collision protein from *E. coli* that does have endonucleolytic activity. The authors then developed a reporter for studying translation rescue in *B. subtilis* and show Alanine-tagged, stalled peptides are produced, but mRNA is not degraded in a MutS2 dependent fashion. A series of MutS2 mutants reveal that in contrast to perturbing MutS2's ATPase activity, mutating the KOW and/or SMR domain the results in disrupted stable binding to collided ribosomes, intermediate amounts of stalled peptide and only mild sensitivity to ribosome stalling antibiotics. Finally in a novel test, they carried out in vitro disome splitting experiments and show that while WT MutS2 with ATP can decrease the disome population and increase 30S/50S subunits, non-hydrolyzable ATP or ATPase mutants cannot. Meanwhile, mutation of the SMR domain (DLR to ALA) still allows for disome splitting. The sum of these results supports the earlier study by Cerullo et al. that MutS2 functions upstream of RqcH to split stalled ribosomes and adds important new evidence that the KOW and SMR domains sense the disome but are not responsible for splitting the ribosomes or cleaving mRNA.

Major points and Questions:

- 1) The discrepancy between the observed domains in the cryo-EM map in this work (SMR and KOW) versus Cerullo et al. (ATPase) is striking but remains unexplained. The current complex is reconstituted with excess MutS2 stalled with ADPNP, while the former work purified a native complex from cells. Is the current state an on-pathway, early state or some off-pathway dead end? The mutations to the KOW and SMR domains and various genetic and biochemical experiments show the found binding positions are relevant for binding. Figure S2 is also very helpful in visualizing how the current and prior structures may be related. But it is unclear to me how the authors drew the conclusion that their structure represents a single homodimer of MutS2. Could the excess amount of MutS2 and ADPNP stalling lead to multiple monomers or dimers to bind non-productively?
- 2) Please define the state (rotated/non-rotated) for the leading and collided ribosomes of the disome in the current structure. There appears to be a difference in the current work (both appear non-rotated) as compared to Cerullo et al. where the collided ribosome was rotated, this difference is not discussed in text and might account for some of the observed differences.
- 3) Following up on the earlier point, what is the multimericity of MutS2 from *B. subtilis*? Is it a constitutive dimer or does the dimer only form on collided ribosomes as part of the sensing mechanism? I do not see this important question addressed in the current work or directly for *B.s.* MutS2 in prior work. Perhaps the size of the dimerization is very large and therefore a high affinity for dimer is assumed or alternatively this question could be addressed by size exclusion chromatography of the currently purified protein.
- 4) A major strength of this paper over prior work is the reconstituted splitting reaction showing purified MutS2 can split stalled "disomes", but some small changes could improve this section.
 - 4.1) What were the disomes purified from? Was it a MutS2 present in the strain/extract? What mRNA sequence were the ribosomes stalled at (non-specifically with an antibiotic or site-specifically with one of the stalling reports?) Details are unclear in the combination of methods section/figure legends/text.
 - 4.2) What is the purity of the purified MutS2 both WT and mutants? No gel is shown for the purified proteins, although the mutations and ATP dependence do suggest that the activity observed is due to MutS2.
 - 4.3) Non-essential but related: Is it clear which of the ribosomes in the disome is getting split? The authors leave this obscure. It is expected to be the stalled ribosome, but is it? Perhaps this point can be addressed by toe printing if the disomes come from a site-specifically stalled construct.

Minor points

- 1) Figure 2B: Consider adding labels to the SMR and KOW domains in this panel as this is the first close up view
- 2) Figure 2F/G: Consider coloring DLR/GxG regions to support callout in the text
- 3) Figure 2F/G: Please define what "hypothetical" *E. coli* SMR binding means, was this a structural alignment? Of which models? Using which features?
- 4) Figure 4B: Is the * band consistent with the peptide on the collided ribosome rather than the stalled ribosome?
- 5) Figure 5A: Please explicitly define how the % disomes is calculated and label the vertical axis. It is unclear if the number reflects area of disomes / (disome+70S) or disomes/(disomes+70S+30S+50S) normalized to starting material. Consider also quantifying increase in subunits.
- 6) Figure 5B: Keep naming for mutants consistent with Figure 4

Non-essential

Have the authors considered placing the non-inhibited (i.e. active) splitting reactions on cryo-EM grids?

Referee #2:

Ribosome quality control mechanisms are essential in all forms of life and include mechanisms to recognize stalled ribosomes via ribosome collisions. The system in *E. coli* responsible for detecting collided ribosomes is not conserved across bacteria, prompting the authors to explore the mechanisms used in other bacteria such as *B. subtilis*. The authors use a combination of bioinformatics, biochemistry, cryo-EM and cell-based experiments to reveal the molecular mechanisms used by the protein MutS2 in *B. subtilis* to detect and clear collided ribosomes stalled on mRNAs. The mechanism of MutS2 differs substantially from that of the *E. coli* protein SmrB. MutS2 seems to act as a dimer and detects stalled ribosomes via its SMR domain, with assistance of its KOW domain. In contrast to *E. coli* SmrB, MutS2 does not cleave the inter-ribosome mRNA, but rather splits the stalled ribosome in an ATP-dependent manner via its ABC ATPase domain into free 30S and peptidyl-tRNA-50S complexes. This is then targeted subsequently by the RQC protein RqcH, which adds poly-Ala to the C-terminus of the aberrant truncated protein to facilitate downstream degradation.

Interestingly, the authors can identify the SMR and KOW domains in their reconstituted samples by cryo-EM, in contrast to Cerullo et al. (ref. 33). By contrast Cerullo et al. see the ATPase domains but not the SMR and KOW domains. There are other differences to note. In Cerullo et al., the stalled ("leading") ribosome lacks ribosomal protein uS2 but recruits protein bS21 which is not a normal component of the *B. subtilis* 30S. By contrast, bS21 is absent from the stalled ribosome in the present work and it is not clear if S2 is missing, as observed in Cerullo et al. (Fig. 2C seems to indicate uS2 is missing or highly flexible and could be viewed at lower resolution.)

Overall, this is a well-written paper, but needs a little more clarity on the differences with the Cerullo et al. work. Do the authors here see uS2 on the stalled ribosome or not? Labels in Fig. 2 would help.

We can clearly identify uS2 on the stalled ribosome both in the *in vitro* complex and the newly added *in vivo* data. As suggested, labels have been added to Fig. 2, as well as a dedicated panel in Fig. EV4 which shows a model-to-map fit of both copies of uS2.

Also, what reasons could explain the difference in bS21 binding? Could the structures represent different steps in the collision and splitting reaction pathway? If not, could the difference be explained by how the samples were prepared? For example, Cerullo et al. purified samples directly from *B. subtilis* cells via a FLAG-tagged MutS2, without using sucrose gradients. In the present work, the authors first purified disomes from *in vitro* translation reactions involving purified *B. subtilis* ribosomes and the remainder coming from the *E. coli* PURE system (refs. 28 and 39) and then added a ten-fold excess of tagged MutS2. It's likely sucrose gradients were used, but it's not clear at what step of the process this would have been done. This should be explicitly included in the methods, as it is a complicated method.

The disomes used in the *in vitro* reconstitution experiments were actually purified by sucrose gradient fractionation from cell-free translation reactions using *B. subtilis* extracts. We apologize that the section detailing the preparation was accidentally omitted from the Methods section. It has been added to the current version.

It is indeed a possibility that the sucrose gradients used for disome preparation resulted in removal of bS21 and thus preclude its later recruitment in the *in vitro* reconstitution reaction. However, we now also show a MutS2-disome complex purified directly from *B. subtilis* cells via FLAG-tagged MutS2 very similar to the preparation shown by Cerullo *et al.* In this complex, we observe density corresponding to the MutS2 SMR domain, but no density occupying the binding pocket of bS21. While the resolution of the reconstruction obtained from these data is limited, it shows clearly that

the MutS2-disome complex which binds the SMR domain in the inter-ribosomal interface does not contain bS21.

Have the authors tried adding back *B. subtilis* bS21 to the *in vitro* translation reactions, to see if it might be bound where they predict a collision based on the present structure? It could be that the SMR domain in Fig. S2 occupies the binding site simply because there is room to fall into that pocket rather than due to excluding bS21. This should be commented on, at the very least.

We have not attempted to add back bS21 since we expect that the cell-free extract contains physiological amounts of this protein anyway. We cannot exclude that adding bS21 back after disome purification when adding MutS2 would have resulted in bS21 association. However, our *in vivo* complex shows that, even in the presence of bS21 in the system, this ribosomal protein is not associated with the MutS2-disome complex when the SMR domain is positioned in the inter-ribosomal interface. As suggested, this is now commented on in the revised text.

Minor: There seems to be some text deleted between the bottom of p. 6 and top of p. 7, possibly hidden behind the figure.

We fixed this error.

Referee #3:

The study by Park et al. dissects the mechanism of a recently identified bacterial ribosome collision sensor named MutS2 from *B. subtilis*. A recent study of this protein (Cerullo et al. *Nature*, 2022) revealed the structure of the homodimeric ATPase domains bound to a collided pair of ribosomes, but did not structurally or biochemically characterize the C-terminal KOW domain and SMR domain, whose fold suggests an endonucleolytic function. Now, Park et al. show that the SMR domain does not cleave mRNA but rather along with KOW contributes weakly to sensing disomes.

Using cryo-EM they visualize the KOW and SMR domains on disomes and study the function of these domains using genetic and biochemical methods. They first purified a stalled disome complex from *B. subtilis* lysates and incubated it with purified MutS2 and in the presence of AMP-PNP and studied the complex via cryo-EM. Cryo-EM maps of disomes reveal two KOW and one SMR domains bound at the interface of the collided ribosomes, while the ATPase domains are not resolved in direct contrast to Cerullo et al. Notably the MutS2 SMR domain at the dimeric interface is positioned very differently from SmrB, a ribosome collision protein from *E. coli* that does have endonucleolytic activity. The authors then developed a reporter for studying translation rescue in *B. subtilis* and show Alanine-tagged, stalled peptides are produced, but mRNA is not degraded in a MutS2 dependent fashion. A series of MutS2 mutants reveal that in contrast to perturbing MutS2's ATPase activity, mutating the KOW and/or SMR domain the results in disrupted stable binding to collided ribosomes, intermediate amounts of stalled peptide and only mild sensitivity to ribosome stalling antibiotics. Finally in a novel test, they carried out *in vitro* disome splitting experiments and show that while WT MutS2 with ATP can decrease the disome population and increase 30S/50S subunits, non-hydrolyzable ATP or ATPase mutants cannot. Meanwhile, mutation of the SMR domain (DLR to ALA) still allows for disome splitting. The sum of these results supports the earlier study by Cerullo et al. that MutS2 functions upstream of RqcH to split stalled ribosomes and adds important new evidence that the KOW and SMR domains sense the disome but are not responsible for splitting the ribosomes or cleaving mRNA.

Major points and Questions:

1) The discrepancy between the observed domains in the cryo-EM map in this work (SMR and KOW) versus Cerullo et al. (ATPase) is striking but remains unexplained. The current complex is reconstituted with excess MutS2 stalled with ADPNP, while the former work purified a native complex from cells. Is the current state an on-pathway, early state or some off-pathway dead end? The mutations to the KOW and SMR domains and various genetic and biochemical experiments show the found binding positions are relevant for binding. Figure S2 is also very helpful in visualizing how the current and prior structures may be related. But it is unclear to me how the authors drew the conclusion that their structure represents a single homodimer of MutS2. Could the excess amount of MutS2 and ADPNP stalling lead to multiple monomers or dimers to bind non-productively?

Based on our structural data, we cannot entirely rule out that the observed state represents an unproductive binding state or off-pathway complex. It is highly likely, however, that it represents an on-pathway (early state) intermediate since we can only observe it upon inhibition of the ATPase activity of MutS2 either by mutation of the ATPase domain or in the presence of a non-hydrolysable ATP analogue. It is not possible to isolate this state in the same way from ATPase active complexes (verified by splitting assays), suggesting that the state we observe can usually progress further to split the stalled ribosome and dissociate. The findings using the *in vivo* complex also support this interpretation.

We agree that we have no direct evidence for the observed densities on the disome belonging to a single homodimer of MutS2. However, the splitting activity observed when using the same cell-free system under the same conditions (with an excess of MutS2) clearly indicates that unproductive binding of multiple monomers or dimers is not prevalent.

2) Please define the state (rotated/non-rotated) for the leading and collided ribosomes of the disome in the current structure. There appears to be a difference in the current work (both appear non-rotated) as compared to Cerullo et al. where the collided ribosome was rotated, this difference is not discussed in text and might account for some of the observed differences. #

We now explicitly discuss the state of the ribosomes in the text and have added a panel in Fig EV4 comparing our *in vivo* data with that of Cerullo *et al.*

In the data obtained from the *in vitro* reconstitution, we found that the MutS2 SMR domain is visible only on disomes in which the collided ribosome is in a non-rotated conformation. In our *in vivo* data, we observed that the collided ribosome in the complex assumes the same rotated conformation described by Cerullo *et al.*

While these findings initially appear contradictory, it is important to keep in mind that ribosomes used for *in vitro* reconstitution no longer exhibit translational activity and therefore are unlikely to undergo further changes in their rotational state that may occur for example after ligand binding *in vivo*.

It is therefore possible that binding of MutS2 *in vivo* may occur also with the collided ribosome preferentially in the non-rotated state as observed *in vitro*, followed by transition of the ribosome to the hybrid state. Notably, this transition reduces the distance between the two collided ribosomes. In contrast, it appears that in the *in vitro* reconstitution only those disomes with the trailing ribosome in the non-rotated state can be bound by MutS2. A rotated conformation of the collided ribosome may disfavor the initial accommodation of the SMR domain due to spatial constraints, but does not exclude its persistence at the ribosomal interface.

3) Following up on the earlier point, what is the multimericity of MutS2 from *B. subtilis*? Is it a constitutive dimer or does the dimer only form on collided ribosomes as part of the sensing mechanism? I do not see this important question addressed in the current work or directly for *B.s. MutS2* in prior work. Perhaps the size of the dimerization is very large and therefore a high affinity for dimer is assumed or alternatively this question could be addressed by size exclusion chromatography of the currently purified protein.

Size exclusion chromatography shows that MutS2 alone runs as a dimer or higher oligomer. These data were added as Fig EV2. MutS2 from other bacteria has been shown previously to form dimers or tetramers *in vitro*, consistent with this observation.

4) A major strength of this paper over prior work is the reconstituted splitting reaction showing purified MutS2 can split stalled "disomes", but some small changes could improve this section.
4.1) What were the disomes purified from? Was it a MutS2 present in the strain/extract? What mRNA sequence were the ribosomes stalled at (non-specifically with an antibiotic or site-specifically with one of the stalling reports?) Details are unclear in the combination of methods section/figure legends/text.

We apologize that the details on the purification of disomes used for these experiments were accidentally left out of the Methods section previously. We have added this section to the current version.

4.2) What is the purity of the purified MutS2 both WT and mutants? No gel is shown for the purified proteins, although the mutations and ATP dependence do suggest that the activity observed is due to MutS2.

We estimate the purity of our preparation to be at least 95%. SDS-PAGE of the purified MutS2 protein has been added to Fig EV2.

4.3) Non-essential but related: Is it clear which of the ribosomes in the disome is getting split? The authors leave this obscure. It is expected to be the stalled ribosome, but is it? Perhaps this point can be addressed by toe printing if the disomes come from a site-specifically stalled construct.

Based on the structure by Cerullo *et al.* showing interaction of the ATPase domains only with the stalled ribosome and also considering other known mechanisms of ribosome rescue (eukaryotic RQT), it is extremely likely that only the stalled ribosome can be split by MutS2. This would be consistent with our findings from the splitting assays, which show an approximately uniform increase in the abundance of both 70S monosomes and ribosomal subunits after splitting. Furthermore, the MS data reveal Ala-tailing of nascent peptides released from the stalled ribosome. Additional verification of this point is unfortunately beyond the scope of this work, which is mainly concerned with the mechanism of recruitment of MutS2 and the role of the SMR and KOW domains.

Minor points

1) Figure 2B: Consider adding labels to the SMR and KOW domains in this panel as this is the first close up view

Done as suggested.

2) Figure 2F/G: Consider coloring DLR/GxG regions to support callout in the text

Done as suggested.

3) Figure 2F/G: Please define what "hypothetical" E. coli SMR binding means, was this a structural alignment? Of which models? Using which features?

Done as suggested in the Figure legend.

4) Figure 4B: Is the * band consistent with the peptide on the collided ribosome rather than the stalled ribosome?

Unfortunately, the design of the reporter used in the MS experiments does not allow us to resolve this question; we would not be able to detect nascent peptides released from the collided ribosome with the current MS approach. It may be that there are a number of possible shorter products. The * band appears in all the lanes, and does not depend on MutS2, the focus of our work here.

5) Figure 5A: Please explicitly define how the % disomes is calculated and label the vertical axis. It is unclear if the number reflects area of disomes / (disome+70S) or disomes / (disomes+70S+30S+50S) normalized to starting material. Consider also quantifying increase in subunits.

Done as suggested

6) Figure 5B: Keep naming for mutants consistent with Figure 4

The names of the mutants were changed so that they match in these figures.

Non-essential

Have the authors considered placing the non-inhibited (i.e. active) splitting reactions on cryo-EM grids?

We made several attempts to vitrify reconstituted active splitting reactions, but were not able to visualize any MutS2-disome complexes from these data.

Dear Dr Buskirk,

Thank you for submitting a revised version of your manuscript. Your study has now been seen by all original referees, who find that their previous concerns have been addressed and now recommend publication of the manuscript. There remain only a few mainly editorial points that have to be addressed before I can extend formal acceptance of the manuscript:

- 1 Please provide source data. For more information please consult the attached PDF.
- 2 Please rename the "conflict of interest" section to "DISCLOSURE AND COMPETING INTERESTS STATEMENT"
- 3 Please convert the Appendix Figures file to PDF and rename to Appendix Figure S1-S2
- 4 Please make sure that the datasets (specific URLs) of the PDB and EMDB datasets are made publicly available latest at the online publication date.
- 5 Please define n in the legend of figure 3f.
- 6 Please define the error bars in the legend of figure 3f.
- 7 Please move the main figure legends between References and EV figure legends.

With best regards,

Cornelius Schneider

Cornelius Schneider, PhD
Editor
The EMBO Journal
c.schneider@embojournal.org

We realize that it is difficult to revise to a specific deadline. In the interest of protecting the conceptual advance provided by the work, we recommend a revision within 3 months (5th Feb 2024). Please discuss the revision progress ahead of this time with the editor if you require more time to complete the revisions. Use the link below to submit your revision:

All editorial and formatting issues were resolved by the authors.

Dear Prof. Buskirk,

I am pleased to inform you that your manuscript has been accepted for publication in the EMBO Journal.

Yours sincerely,

Cornelius Schneider, PhD
Editor
The EMBO Journal
c.schneider@embojournal.org
